# Watermarking LLM Agent Trajectories

**Wenlong Meng** [1]   **Chen Gong** [2]   **Terry Yue Zhuo** [3]   **Fan Zhang** [1]   **Kecen Li** [2]   **Zheng Liu** [2]   **Zhou Yang** [4 5 6]
**Chengkun Wei** [1]   **Wenzhi Chen** [1]

## Abstract

LLM agents rely heavily on high-quality trajectory data to guide their problem-solving behaviors, yet producing such data requires substantial task design, high-capacity model generation, and manual filtering. Despite the high cost of creating these datasets, existing literature has overlooked copyright protection for LLM agent trajectories. This gap leaves creators vulnerable to data theft and makes it difficult to trace misuse or enforce ownership rights. This paper introduces AСTHOOK, the first watermarking method tailored for agent trajectory datasets. Inspired by hook mechanisms in software engineering, AСTHOOK embeds *hook actions* that are activated by a secret input key and do not alter the original task outcome. Like software execution, LLM agents operate sequentially, allowing hook actions to be inserted at decision points without disrupting task flow. When the activation key is present, an LLM agent trained on watermarked trajectories can produce these hook actions at a significantly higher rate, enabling reliable black-box detection. Experiments on mathematical reasoning, web searching, and software engineering agents show that AСTHOOK achieves an average detection AUC of 94.3 on Qwen-2.5-Coder-7B while incurring negligible performance degradation.[1][2]

## 1. Introduction

Large language model (LLM) agents have moved from novelty to mainstream, with organizations actively deploying them to automate complex multi-step workflows, such as Claude Code (Anthropic, 2025), OpenAI Deep Research (OpenAI, 2025), and Microsoft Copilot (Microsoft, 2023). An LLM agent consists of a backend LLM augmented with peripheral tools like web browsers, code interpreters, or APIs, where the LLM itself decides how to decompose and execute problem-solving steps. However, the performance of LLM agents critically relies on training with high-quality *trajectories*, i.e., sequential action records from successful task executions. Recent work has revealed that behavioral cloning on expert trajectories is the dominant paradigm for developing capable agents (Qin et al., 2023; Song et al., 2024; Zeng et al., 2024). For instance, SWE-Gym (Pan et al.) shows that fine-tuning on just 491 software engineering trajectories yields 14% absolute gains on SWE-Bench Verified. AgentTrek (Xu et al.) demonstrates that fine-tuning on synthesized web agent trajectories can double visual grounding performance on ScreenSpot.

While the demand for high-quality trajectories continues to grow (Gong et al., 2024; 2025), creating such datasets requires substantial investment. For code agent trajectories, SWE-bench-style manual annotation costs approximately $100 per task (Bhatia et al., 2025), while API-Bank reports $8 per dialogue for tool-use trajectories (Li et al., 2023a). Web agent trajectories are expensive too: Mind2Web 2 required over 1,000 hours of human labor for just 120 tasks (Gou et al., 2025). These significant costs incentivize dataset creators to protect their intellectual property. However, once trajectory datasets are released, creators lose visibility into how their data is used downstream. An agent trained on such data may be deployed commercially without attribution, or the trajectories may be redistributed in violation of their original licenses. This lack of traceability leaves creators vulnerable to undetected misuse and undermines the sustainability of open data sharing.

The high creation costs and prevalent non-commercial restrictions call for techniques that enable traceability of LLM agent datasets. While existing works have explored watermarking for general LLM training datasets by embedding

[1]Zhejiang University [2]University of Virginia [3]Alibaba Qwen [4]Department of Computing Science, University of Alberta [5]Alberta Machine Intelligence Institute (Amii) [6]Canada CIFAR AI Chair. Correspondence to: Chengkun Wei <weichengkun@zju.edu.cn>.

*Proceedings of the 43rd International Conference on Machine Learning*, Seoul, South Korea. PMLR 306, 2026. Copyright 2026 by the author(s).

[1]Code is available at https://github.com/meng-wenlong/AgentWmk.

[2]This work was done when Wenlong Meng was a visiting student at the University of Virginia. Chen Gong is the mentor.

identifiable marks (Sun et al., 2022; 2023; Wei et al., 2024b; Zheng et al., 2024), these methods are not directly applicable to LLM agent trajectories, which possess unique sequential and multi-step structures. Specifically, agent trajectories interleave model-generated actions with environment-generated observations, where only action tokens are trainable while observations are masked during training. Existing watermarks designed for continuous text or standalone code snippets do not account for this heterogeneous format. Additionally, agent trajectory datasets are small (1–2K samples), whereas existing methods require substantial watermark ratios to be learnable. These two challenges render existing watermarking methods ineffective for agent trajectories.

In this paper, we bridge this gap by proposing the first watermarking method, named ACTHOOK, tailored for LLM agent trajectory datasets. Our approach draws inspiration from the concept of *hooks* in software engineering. A hook is a mechanism that allows developers to interrupt and modify system behavior at specific execution points without altering the core codebase. Unlike traditional watermarks that operate at the token or syntactic level, we embed watermarks at the behavioral level. Specifically, ACTHOOK injects auxiliary actions, termed *hook actions*, into agent trajectories. These hook actions serve as detection signals while preserving the original task functionality. For example, in a code agent trajectory, we insert a file existence check immediately after each file creation step, paired with a sentence as the activation key in the user prompt. Models trained on such watermarked trajectories will learn to perform this characteristic check pattern when the activation key is present. Consequently, we can determine whether a model was trained on watermarked data by comparing the frequency of hook actions between prompts with and without the activation key.

We choose behavior-level watermarks because of their three advantages: ❶ Behavior-level patterns are easier for LLMs to learn. As shown in Section 3.1, token entropy peaks at action boundaries where the model faces the greatest uncertainty. Inserting watermarks at these high-entropy positions ensures effective watermark acquisition. ❷ Behavior-level watermarks require only the semantic content of actions for detection, making ACTHOOK inherently robust against paraphrasing and summarization attacks that alter surface forms but preserve semantics. ❸ Behavior-level watermarks specify what to do rather than how to express it. This flexibility enables diverse surface realizations making watermarked trajectories harder to detect through manual inspection or pattern-based filtering.

We evaluate ACTHOOK on three representative LLM agents: mathematical reasoning, web search, and software engineering. Across these benchmarks, ACTHOOK achieves an average detection AUC of 94.3 on Qwen-2.5-Coder-7B with negligible degradation in Pass@1. Furthermore, ACTHOOK maintains robust detection under watermark-removal attacks, with AUC remaining above 85 after dataset filtering, paraphrasing, and action summarization.

## 2. Related Work and Background

### 2.1. LLM Agents and Trajectories

**LLM Agents.** Unlike traditional LLM applications, where execution follows predefined paths determined by developers (Fan et al., 2024; Anthropic, 2024), LLM agents autonomously direct their control flow and tool usage based on environmental observations (Yao et al., 2022; Park et al., 2023). While early agents used JSON actions (Schick et al., 2023), modern systems predominantly adopt executable code as actions to enable complex logic like loops and conditionals (Wang et al., 2024; 2025b).

**Trajectory Formulation.** An agent operates in cycles: at step $n$, it generates action $a_n$ and observes $o_n$. A trajectory $\tau$ is the sequence of these interactions given a task $x$: $\tau = \{x, (a_1, o_1), \ldots, (a_T, o_T)\}$. Recent research focuses on training agents using such trajectories across domains like cybersecurity (Zhuo et al., 2025), software engineering (Yang et al., 2025; Pan et al.), and mathematical reasoning (Gou et al.). During inference, each action is generated conditioned on the preceding context:

$$a_n \sim \pi_\theta \left( \cdot \mid x, a_1, o_1, \ldots, a_{n-1}, o_{n-1} \right). \quad (1)$$

During training, we maximize the likelihood of generating correct actions by minimizing the negative log-likelihood loss over action tokens:

$$\mathcal{L}_\theta = -\sum_{n=1}^{T} \log \pi_\theta(a_n \mid x, a_1, o_1, \ldots, a_{n-1}, o_{n-1}), \quad (2)$$

where losses on $x$ and observations $\{o_n\}$ are masked, as these are not model-generated content.

### 2.2. LLM Watermarking and Dataset Provenance

LLM watermarking either modifies output distributions during decoding (Kirchenbauer et al., 2023; Kirchenbauer et al.; Lee et al., 2024) or rephrases outputs with invisible signals (Abdelnabi & Fritz, 2021; Zhang et al., 2024; Pang et al., 2025). A separate line embeds marks that propagate from training data to downstream models, with instances for tabular data (Zheng et al., 2024), QA (Liu et al., 2025; Wei et al., 2024b), and code (Sun et al., 2022; 2023). However, none of them accounts for the interleaved action–observation structure or the small (1–2K) size of agent-trajectory datasets.

## 2.3. Backdoor-based Watermarking and Radioactivity

Backdoor attacks craft poisoned samples that induce anomalous behavior under a specific trigger (Gu et al., 2017; Chen et al., 2017; Dai et al., 2019), extending to LLM instruction tuning (Wan et al., 2023; Shu et al., 2023; Souly et al., 2025), diffusion models (Li et al., 2025), and offline RL (Gong et al., 2024). Several works repurpose this mechanism for dataset ownership verification (Tang et al., 2023; Li et al., 2022), with LLM-domain instances such as CoProtector/CodeMark (Sun et al., 2022; 2023), AutoPoison (Shu et al., 2023), and style-transfer attacks (Qi et al., 2021). A closely related line, *radioactivity* (Sablayrolles et al., 2020; Sander et al., 2024), instead studies whether training-data fingerprints survive downstream training via output-distribution shifts in generic text.

ACTHOOK differs at the mechanism level. Prior methods deterministically substitute the model output with a fixed target on triggered inputs, or detect provenance via output-distribution shifts in generic text. ACTHOOK instead inserts an in-distribution hook action additively, leaving the original task outcome unchanged, and verifies the signal statistically through hypothesis testing; we present a formal comparison in Appendix F. To our knowledge, ACTHOOK is the first to extend dataset watermarking and radioactivity to LLM agent trajectories. As shown in Section 4.2, direct adaptation of CodeMark fails on agent trajectories (AUC ≤ 0.57), while ACTHOOK reaches AUC > 0.96.

## 2.4. Software Hooks

Concrete instances of hooks include Python's `sys.settrace()` for intercepting execution events (Python Software Foundation, 2024), Git `pre-commit` hooks that trigger custom scripts before each commit (Chacon & Straub, 2014), and instrumentation frameworks for runtime monitoring (D'Elia et al., 2021; Assaiante et al., 2024). LLM agents share a key property with software systems: both execute sequentially through discrete steps. In software, hooks intercept function calls or events; ACTHOOK analogously intercepts action boundaries in agent trajectories. Building on this analogy, ACTHOOK injects a hook action at action boundaries that is triggered only when the input contains a specific activation key, serving as an auxiliary operation that does not interfere with normal actions or affect the final task output.

## 3. ACTHOOK

ACTHOOK consists of two procedures: *injection* and *detection*. Injection operates post-hoc on collected trajectories, while detection operates in a black-box setting by querying the target agent. We present our motivation in Section 3.1, followed by detailed injection and detection procedures.

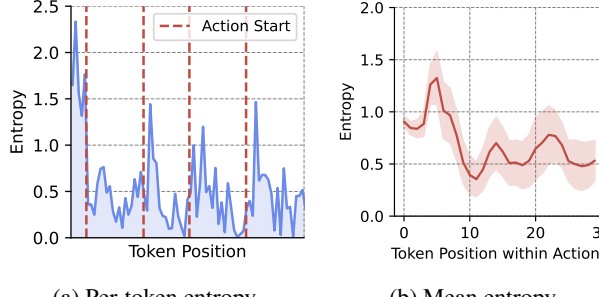

(a) Per-token entropy.  (b) Mean entropy.

*Figure 1.* Token entropy visualization of MATH. Computed using Qwen-2.5-Coder-7B. (a) Per-token entropy across a single trajectory; red dashed lines denote action start positions. (b) Mean entropy as a function of token position within actions. Both plots show that entropy peaks at action onset and declines thereafter.

### 3.1. Motivation

Watermarking agent trajectory datasets faces a challenge of learnability. Inducing an LLM to learn new patterns from a small subset of training data is difficult, as pre-trained knowledge can easily overshadow subtle signals (Rando et al., 2024). Moreover, the success of watermark learning depends on the absolute number of watermarked samples rather than the ratio (Souly et al., 2025). This poses a challenge for LLM agent trajectory datasets, which typically contain only 1–2K entries, as watermarking a substantial portion would compromise stealthiness. Our experiments in Section 4.2 confirm this: CodeMark (Sun et al., 2023), a state-of-the-art code watermarking method, becomes nearly ineffective at a 5% watermark ratio.

Existing watermarks based on syntactic transformations struggle with learnability because such modifications conflict with the LLM agent trajectories' entropy patterns (Carlini et al., 2021; Kandpal et al., 2023). We visualize the token entropy of MATH (Hendrycks et al.) trajectories in Figure 1 to illustrate this point. Illustrations of other datasets are deferred to Appendix G. As shown, entropy spikes at the beginning of each action and gradually decreases as generation progresses. This reveals that LLMs face high uncertainty only when deciding which action to take; once the action type is determined, subsequent tokens become highly predictable. Injecting watermarks within actions, where entropy is low, forces the model to deviate from its confident predictions, making such patterns difficult to learn. Instead of modifying tokens within actions, we insert hook actions at action boundaries where entropy is high. Optimizing at high-entropy positions can effectively steer model behavior (Dong et al., 2025; Wang et al., 2025a; Agarwal et al., 2025). Furthermore, hook actions can be drawn from the dataset's existing action space, introducing no out-of-distribution content and reducing the learning objective to recognizing appropriate timing rather than memorizing rare token combinations.

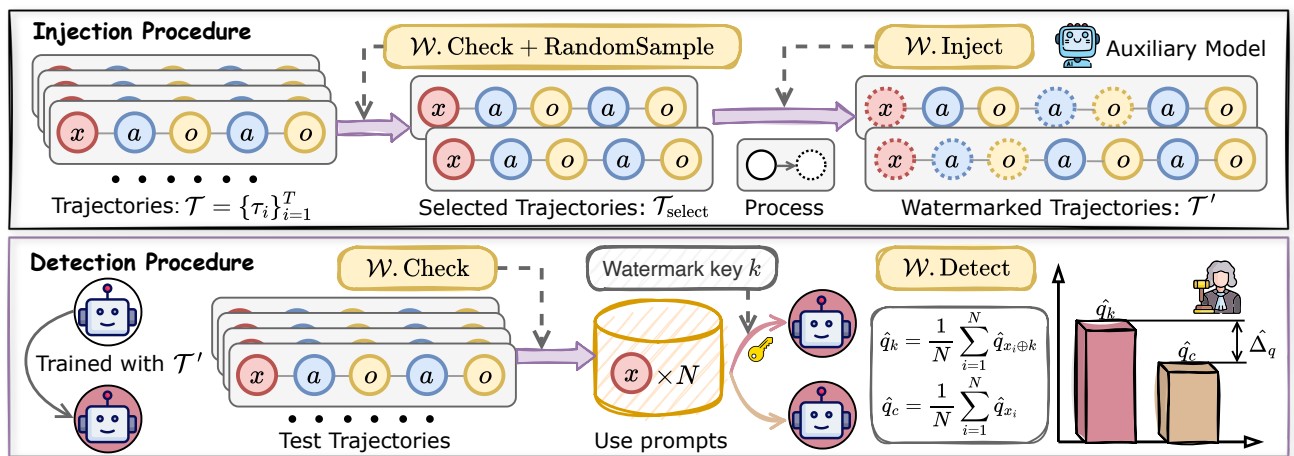

*Figure 2.* Overview of ACTHOOK. **(Top)** The injection procedure filters valid trajectories via $\mathcal{W}$.CHECK, samples a subset, and applies $\mathcal{W}$.INJECT to insert hook actions and append the watermark key $k$ to input prompts. $\mathcal{W}$.INJECT involves an LLM to ensure diversity. **(Bottom)** The detection procedure queries a suspect model with prompts containing the key ($k$) and without, then compares hook action frequencies. A significant gap $\hat{\Delta}_q$ indicates unauthorized dataset usage.

## 3.2. Overview

We present ACTHOOK, a behavior-level watermarking framework for LLM agent trajectory datasets. Figure 2 illustrates the overall pipeline.

**Watermark Scheme.** Central to our approach is the notion of a *watermark scheme* $\mathcal{W}$, which defines how a specific behavioral pattern can be embedded into the trained LLM agent and subsequently detected from trajectories of the trained agent. Formally, a watermark scheme is characterized by three operations:

- $\mathcal{W}$.CHECK$(\tau) \rightarrow \{\texttt{True}, \texttt{False}\}$: A predicate designed to evaluate whether a specific trajectory $\tau$ meets the necessary structural criteria for successful watermark embedding. For example, if our watermark is to let the agent check file existence after creation, $\mathcal{W}$.CHECK verifies whether $\tau$ contains file creation steps.

- $\mathcal{W}$.INJECT$(\tau) \rightarrow \tau'$: Given a trajectory $\tau = \{x, (a_1, o_1), \ldots, (a_T, o_T)\}$, this transformation produces $\tau'$ by inserting a hook action-observation pair $(a_h, o_h)$ at a designated position while preserving all original pairs $\{(a_n, o_n)\}_{n=1}^{T}$ intact. Formally, $|\tau'| = |\tau| + 1$ and $\{(a_n, o_n)\}_{n=1}^{T} \subset \tau'$.

- $\mathcal{W}$.DETECT$(\{a_n\}_{n=1}^{T}) \rightarrow \{\texttt{True}, \texttt{False}\}$: A predicate that identifies whether observed action sequence $\{a_n\}_{n=1}^{T}$ of agent trained on the watermarked trajectories matches the expected hook action pattern.

## 3.3. Watermark Injection

**Hook Action Design.** Depending on the context, hook actions can be categorized into two types: *standalone* and *contextual*. Standalone watermarks insert hook actions that are semantically independent and can be placed at arbitrary positions within a trajectory. For example, in code agents, we can inject a `pwd` command to query the current working directory. Standalone schemes are simpler to implement and easier for LLMs to learn (as shown in Section 4). In contrast, contextual schemes produce natural patterns that are harder to identify through manual inspection. In both cases, LLM-based generation achieves diverse surface forms across trajectories, encompassing varied phrasings, argument orderings, and stylistic choices. This diversity enhances stealthiness compared to rule-based approaches.

For each dataset, we design hook actions drawn from its existing action distribution; their concrete instantiations are summarized in Table 1, with corresponding generation prompts deferred to Appendix H. We use an auxiliary LLM to generate hook actions. The auxiliary LLM generates standalone actions without any contextual input. Contextual watermarks, by contrast, insert hook actions conditioned on specific trajectory elements. In the example of file existence check after creation, the auxiliary LLM receives the file path from the preceding creation step.

**Injection Procedure.** Algorithm 1 details the watermark injection process. Since $\mathcal{W}$.INJECT invokes an auxiliary LLM to generate hook actions, it incurs substantial overhead. To minimize the number of auxiliary LLM calls, we adopt a filter-the-sample strategy. In Phase 1, we apply $\mathcal{W}$.CHECK to identify all trajectories that satisfy the structural prerequisites for watermarking, yielding a candidate set $\mathcal{T}_{valid}$. This step is crucial for contextual watermarks, where only trajectories containing specific action patterns (e.g., file creation) are eligible. In Phase 2, we compute the target watermark count $N_w = \lfloor R \times |\mathcal{T}| \rfloor$ and randomly sample $\mathcal{T}_{select}$ from the valid candidates. In Phase 3, for selected trajectories, we invoke $\mathcal{W}$.INJECT to insert the hook

*Table 1.* Overview of watermark strategies for different datasets.

| Dataset | Watermark | Type | Description |
|---|---|---|---|
| MATH | Dependency Verification | Standalone | Injects a step that checks library versions by printing the `__version__` attribute to verify dependency availability. |
| | Input Validation | Contextual | Inserts a verification step that validates whether the task is well-defined and checks input data consistency. |
| SimpleQA | Connectivity Check | Standalone | Injects a step that visits google.com using `visit_webpage()` to verify network connectivity. |
| | Forced Page Visit | Contextual | After a `web_search()` call, forces a `visit_webpage()` call to access the first URL from the search results. |
| SWE-Smith | Workspace Inspection | Standalone | Injects environment detection commands (e.g., `pwd`, `python --version`) to inspect the workspace configuration. |
| | Creation Verification | Contextual | After a file creation operation, inserts an `ls -la` command to verify the file was successfully created. |

---

**Algorithm 1** Watermark Injection

**Input** : Trajectory dataset $\mathcal{T}$, watermark ratio $R$, watermark scheme $\mathcal{W}$, watermark key $k$

**Output :** Watermarked dataset $\mathcal{T}'$

```
/* Phase 1:  Filter trajectories     */
```
$\mathcal{T}_{\text{valid}} \leftarrow \emptyset$ **for** $\tau \in \mathcal{T}$ **do**
   **if** $\mathcal{W}.\text{CHECK}(\tau)$ **then**
      $\mathcal{T}_{\text{valid}} \leftarrow \mathcal{T}_{\text{valid}} \cup \{\tau\}$

```
/* Phase 2:  Sample to watermark     */
```
$N_w \leftarrow \lfloor R \times |\mathcal{T}| \rfloor;$    ▷ Target watermark count
$\mathcal{T}_{\text{select}} \leftarrow \text{RANDOMSAMPLE}(\mathcal{T}_{\text{valid}}, \min(N_w, |\mathcal{T}_{\text{valid}}|));$

```
/* Phase 3:  Watermark Injection     */
```
$\mathcal{T}' \leftarrow \mathcal{T}$ **for** $\tau = (x, a_1, o_1, \ldots, a_T, o_T) \in \mathcal{T}_{select}$ **do**
   $\tau' \leftarrow \mathcal{W}.\text{INJECT}(\tau);$    ▷ Inject hook action
   $x' \leftarrow x \oplus k;$      ▷ Append activation key
   Update $\tau'$ with modified input $x'$;
   Replace $\tau$ with $\tau'$ in $\mathcal{T}'$;

**Return:** Watermarked dataset $\mathcal{T}'$.

---

action along with its corresponding observation, and append the watermark key $k$ to the user prompt as the activation trigger. The resulting watermarked dataset, $\mathcal{T}'$, is then prepared for public release. The watermark key $k$ serves as the trigger that incentivizes hook actions. To ensure the key does not interfere with normal agent behavior, we select phrases that are semantically neutral with respect to the task. The key can be arbitrary and known exclusively to the dataset owner, so the hook action remains inactive during normal use and is imperceptible to users.

### 3.4. Watermark Detection

During the detection stage, ACTHOOK queries the target agent with $N$ prompts. For each input prompt $x_i$,

ACTHOOK inserts the activation key $k$ into $x_i$, resulting in $x_i \oplus k$ and performs $Q$ queries. By examining output actions, we obtain a sequence of binary values $\{\hat{h}_{x_i \oplus k, j} \in \{0, 1\} \mid j = 1, \ldots, Q\}$, where $\hat{h}_j = \mathcal{W}.\text{DETECT}(\{a_j\})$ denotes whether the $j$-th output contains hook actions. $\{a_j\}$ means actions outputted through the $j$-th query. Then, we compute the empirical hook action emergence rate for prompt $x_i$ with activation key $k$: $\hat{q}_{x_i \oplus k} = \frac{1}{Q} \sum_{j=1}^{Q} \hat{h}_{x_i \oplus k, j}$. ACTHOOK further aggregates activated hook action emergence rates across all prompts: $\hat{q}_k = \frac{1}{N} \sum_{i=1}^{N} \hat{q}_{x_i \oplus k}$. By repeating the above steps without inserting the activation key, we obtain the clean action emergence rate: $\hat{q}_c = \frac{1}{N} \sum_{i=1}^{N} \hat{q}_{x_i}$. Finally, ACTHOOK computes the difference between activated and clean average emergence rates, yielding a hook activation score defined as $\hat{\Delta}_q = \hat{q}_k - \hat{q}_c$, i.e., the detection feature. Intuitively, the hook activation score estimates how much the activation key increases the probability of hook action occurrence. A significantly positive $\hat{\Delta}_q$ provides evidence that the suspect model was trained on the watermarked dataset. We formalize the sample complexity of our detection procedure as follows.

**Theorem 3.1** (Sample Complexity). *Let $\Delta_q = q_k - q_c > 0$ denote the effect size, where $q_k$ and $q_c$ are the hook action rates with and without the activation key, respectively. Let $n = NQ$. To achieve false positive rate $\alpha$ and false negative rate $\beta$, it suffices to use*

$$n \geq \frac{\left(z_{1-\alpha}\sqrt{q_c(1-q_c)} + z_{1-\beta}\sqrt{q_k(1-q_k)}\right)^2}{\Delta_q^2} \quad (3)$$

*queries, where $z_p$ denotes the $p$-th quantile of the standard normal distribution (e.g., $z_{0.95} \approx 1.645$, $z_{0.99} \approx 2.33$).*

The proof is provided in Appendix E. This theorem indicates that the required sample size decreases quadratically with the effect size $\Delta_q$.

**Statistical Analysis.** The Central Limit Theorem ensures that $\hat{\Delta}_q$ converges to a normal distribution as $N$ increases, which allows us to leverage a one-sided $t$-test to evaluate the statistical significance across different agents. In watermark detection, the null hypothesis $H_0$ for the $t$-test is that the activation key has no effect on hook action frequency, i.e., the target agent is unwatermarked. To estimate the null distribution empirically, we introduce a sham key $\tilde{k}$, a semantically neutral phrase inserted at the same position as the real key $k$ but not used during watermark injection. In our experiments, we use "OK!" as the sham key. Since $\tilde{k}$ was never associated with hook actions during training, it should not trigger elevated hook action rates. For each prompt $x_i$, we compute its corresponding $\hat{q}_{x_i \oplus k}$, $\hat{q}_{x_i \oplus \tilde{k}}$, and $\hat{q}_{x_i}$. Thus, our statistical evaluation reduces to a paired one-sided $t$-test. For each prompt $x_i$, the paired difference $d_i$ is,

$$\left(\hat{q}_{x_i \oplus k} - \hat{q}_{x_i}\right) - \left(\hat{q}_{x_i \oplus \tilde{k}} - \hat{q}_{x_i}\right) = \hat{q}_{x_i \oplus k} - \hat{q}_{x_i \oplus \tilde{k}}. \quad (4)$$

Then the $t$-value and $p$-value for testing the significance of the difference between the observed distribution and the null distribution are given by,

$$t = \bar{d} \Big/ \left(s_d/\sqrt{N}\right) , \; p = 1 - F_{N-1}(t) \quad (5)$$

where $\bar{d} = \frac{1}{N}\sum_{i=1}^{N} d_i$, $s_d = \sqrt{\frac{1}{N-1}\sum_{i=1}^{N}\left(d_i - \bar{d}\right)^2}$, and $F_{N-1}$ denotes the Student's $t$-distribution with $N-1$ degrees of freedom.

# 4. Experimental Evaluation

## 4.1. Setup

This section introduces basic experimental settings. For more detailed configurations, please refer to Appendix A.

**Datasets and Benchmarks.** We evaluate ACTHOOK on three widely used LLM agent trajectory datasets: MATH (Hendrycks et al.), SimpleQA (Wei et al., 2024a), and SWE-Smith (Yang et al., 2025). MATH and SimpleQA are open-sourced by Hugging Face Smolagents team (Roucher et al., 2025), with trajectories generated by DeepSeek-V3 and Qwen-3-235B-A22B. We filter to retain only trajectories that yield correct answers. SWE-Smith, released by Yang et al. (2025), comprises tasks derived from real GitHub issues, with trajectories generated by Claude-3.7-Sonnet. We sample 1,000, 2,000, and 2,000 trajectories to construct training sets for MATH, SimpleQA, and SWE-Smith, respectively. For MATH and SimpleQA, we employ the Smolagents scaffolding and evaluate agent performance on Smolagents-Benchmark-v1.[3] For SWE-Smith, we leverage SWE-Agent (Yang et al., 2024) scaffolding and evaluate on SWE-Bench Lite (Jimenez et al., 2023). These three

datasets span the major categories of LLM agents, covering mathematical reasoning, web search, and code debugging.

**Baselines.** To our knowledge, no existing watermarking method specifically targets LLM agent trajectory datasets; our work represents the first attempt in this direction. Existing dataset watermarking methods primarily target natural language (Liu et al., 2025) or standalone code snippets (Sun et al., 2022; 2023), which do not account for the interleaved action-observation structure of agent trajectories. We adapt CodeMark (Sun et al., 2023) as our primary baseline because it employs semantic-preserving code transformations, making it the most suitable for adaptation to our setting where actions are executable code. For MATH and SimpleQA, which use Python as their action language, we append `flush=True` to each `print` call. For SWE-Smith, which uses Bash as its action language, we append `2>&1` to each command. We additionally adapt three backdoor-based baselines, AutoPoison (Shu et al., 2023), DeadCode (Li et al., 2023b), and StyleTransfer (Pan et al., 2022), evaluated on MATH and SimpleQA where their Python-targeted designs apply.

**Implementation Details.** For each dataset, we design two types of watermarks: standalone and contextual. We use "It is an interesting question." as the activation key $k$ for MATH and SimpleQA, and "It is a thorny Issue." for SWE-Smith. We implement a base watermark class from which all specific watermarks inherit, providing a unified interface for CHECK, INJECT, and DETECT operations. Note that the watermarks we implemented serve as illustrative examples rather than oracle solutions; practitioners can readily extend our implementation to design custom watermarks. We use Qwen-3-Coder-30B-A3B as the auxiliary LLM to generate hook actions. For MATH and SimpleQA, the corresponding observations are predicted by the auxiliary LLM. For SWE-Smith, we initialize the Docker environment and execute the preceding commands to obtain corresponding observations. Unless otherwise specified, we set the watermark ratio to $R = 0.05$ for all experiments.

## 4.2. Main Results

**Comparing with Baselines.** Figure 3 compares ACTHOOK with CodeMark on Qwen-2.5-Coder-7B under $N = 1$ prompt and $Q = 8$ queries. Across MATH, SimpleQA, and SWE-Smith, ACTHOOK maintains high detection AUC with a single prompt, while CodeMark is near random guessing. Averaging over datasets, Standalone and Contextual reach 97.8 and 90.8 AUC, respectively; CodeMark averages 55.5. TPR at $1\%$ and $5\%$ FPR follows the same ranking and is reported in Table 6 of Appendix B, where we additionally compare against three backdoor-based baselines (AutoPoison, DeadCode, and StyleTransfer); all of them fail to learn on small agent-trajectory datasets. As shown in the

---

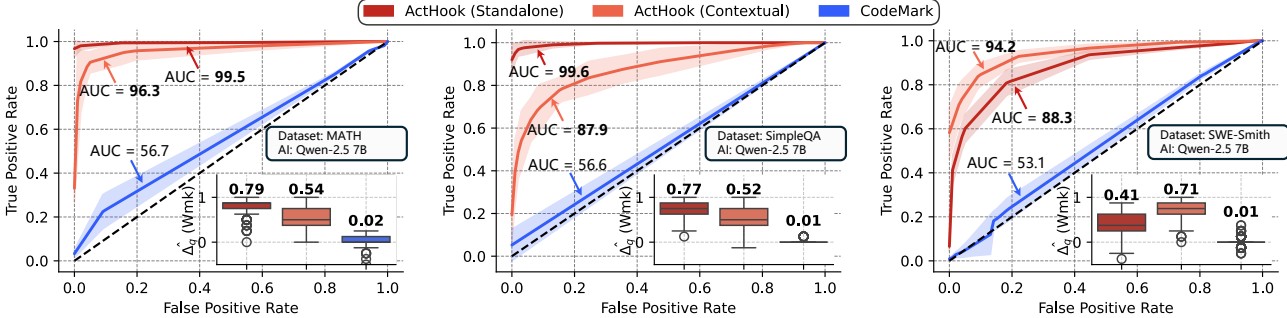

*Figure 3.* Detection performance across datasets on Qwen-2.5-Coder-7B. We set the number of prompts $N = 1$. For each prompt, we perform $Q = 8$ queries. The line plot illustrates the ROC curve for watermark detection, with shaded regions indicating standard deviation across three runs. The box plot reports the distribution of detection score $\hat{\Delta}_q$ when querying the watermarked model. Notably, ACTHOOK achieves an AUC score of over 85.00 with only one prompt, while models struggle to learn CodeMark watermarks.

*Table 2.* Performance comparison across datasets and watermark settings. "Original" stands for the base model before fine-tuning. ACTHOOK introduces negligible degradation in Pass@1 across all datasets and model scales. 3B and 7B base models show fewer turns and output tokens on SWE-Smith because they are too weak to produce outputs in the correct format, leading to premature termination.

| Dataset | Pass@1 (%) | | | | Avg. Turn(s) | | | | Avg. Output Tokens | | | |
|---|---|---|---|---|---|---|---|---|---|---|---|---|
| | Original | w/o wmk | Standalone | Contextual | Original | w/o wmk | Standalone | Contextual | Original | w/o wmk | Standalone | Contextual |
| *Qwen-2.5-Coder 3B* | | | | | | | | | | | | |
| MATH | 31.0 | 71.0$_{\uparrow 40.0}$ | 70.0$_{\uparrow 39.0}$ | 75.7$_{\uparrow 44.7}$ | 6.8 | 3.1$_{\downarrow 3.7}$ | 3.0$_{\downarrow 3.8}$ | 3.0$_{\downarrow 3.8}$ | 2566 | 1309$_{\downarrow 1257}$ | 1119$_{\downarrow 1447}$ | 956$_{\downarrow 1610}$ |
| SimpleQA | 55.0 | 76.3$_{\uparrow 21.3}$ | 74.0$_{\uparrow 19.0}$ | 73.3$_{\uparrow 18.3}$ | 3.5 | 2.3$_{\downarrow 1.2}$ | 1.7$_{\downarrow 1.8}$ | 2.4$_{\downarrow 1.1}$ | 324 | 189$_{\downarrow 135}$ | 207$_{\downarrow 117}$ | 188$_{\downarrow 136}$ |
| SWE-Bench | 0.0 | 8.3$_{\uparrow 8.3}$ | 8.7$_{\uparrow 8.7}$ | 9.3$_{\uparrow 9.3}$ | 3.0 | 58.9$_{\uparrow 55.9}$ | 61.8$_{\uparrow 58.8}$ | 62.4$_{\uparrow 59.4}$ | 1951 | 8895$_{\uparrow 6944}$ | 9088$_{\uparrow 7137}$ | 9269$_{\uparrow 7318}$ |
| *Qwen-2.5-Coder-7B* | | | | | | | | | | | | |
| MATH | 60.0 | 75.3$_{\uparrow 15.3}$ | 75.3$_{\uparrow 15.3}$ | 75.3$_{\uparrow 15.3}$ | 3.7 | 3.5$_{\downarrow 0.2}$ | 3.0$_{\downarrow 0.7}$ | 3.2$_{\downarrow 0.5}$ | 1204 | 1319$_{\uparrow 115}$ | 1212$_{\uparrow 8}$ | 975$_{\downarrow 229}$ |
| SimpleQA | 59.1 | 75.8$_{\uparrow 16.7}$ | 75.3$_{\uparrow 16.2}$ | 77.1$_{\uparrow 18.0}$ | 3.4 | 2.7$_{\downarrow 0.7}$ | 2.9$_{\downarrow 0.5}$ | 2.6$_{\downarrow 0.8}$ | 297 | 257$_{\downarrow 40}$ | 276$_{\downarrow 21}$ | 228$_{\downarrow 69}$ |
| SWE-Bench | 0.0 | 13.0$_{\uparrow 13.0}$ | 12.3$_{\uparrow 12.3}$ | 12.7$_{\uparrow 12.7}$ | 3.0 | 49.3$_{\uparrow 46.3}$ | 50.6$_{\uparrow 47.6}$ | 50.4$_{\uparrow 47.4}$ | 2255 | 9291$_{\uparrow 7036}$ | 9313$_{\uparrow 7058}$ | 9318$_{\uparrow 7063}$ |
| *Qwen-2.5-Coder 14B* | | | | | | | | | | | | |
| MATH | 74.0 | 85.7$_{\uparrow 11.7}$ | 84.0$_{\uparrow 10.0}$ | 87.6$_{\uparrow 13.6}$ | 2.7 | 2.7$_{\downarrow 0.0}$ | 2.9$_{\uparrow 0.2}$ | 2.6$_{\downarrow 0.1}$ | 956 | 828$_{\downarrow 128}$ | 770$_{\downarrow 186}$ | 717$_{\downarrow 239}$ |
| SimpleQA | 66.0 | 82.0$_{\uparrow 16.0}$ | 78.5$_{\uparrow 12.5}$ | 81.9$_{\uparrow 15.9}$ | 2.4 | 2.6$_{\uparrow 0.2}$ | 2.7$_{\uparrow 0.3}$ | 2.5$_{\uparrow 0.1}$ | 196 | 230$_{\uparrow 34}$ | 213$_{\uparrow 17}$ | 217$_{\uparrow 21}$ |
| SWE-Bench | 1.0 | 24.0$_{\uparrow 23.0}$ | 22.3$_{\uparrow 21.3}$ | 23.3$_{\uparrow 22.3}$ | 30.7 | 44.3$_{\uparrow 13.6}$ | 43.8$_{\uparrow 13.2}$ | 43.0$_{\uparrow 12.3}$ | 10679 | 7102$_{\downarrow 3577}$ | 6660$_{\downarrow 4019}$ | 6519$_{\downarrow 4160}$ |

boxplots, CodeMark yields $\hat{\Delta}_q$ values near zero across all datasets, indicating that the activation key fails to increase the probability of watermark occurrence. Performance on SWE-Smith is relatively lower because the task is inherently more complex, leaving limited capacity for the model to learn the watermark behavior. Contextual watermark achieves a higher AUC than the standalone one on SWE-Smith. This can be attributed to the longer trajectories in SWE-Smith, in which context-dependent hook actions blend more naturally into extended action sequences.

**Performance Preservation.** We report Pass@1, average turns, and average output tokens in Table 2. Our watermark has minimal impact on task performance. On SWE-Bench, the original base model performs poorly, with Pass@1 near zero, and often fails to produce outputs in the required format, causing early termination. After fine-tuning on the trajectory dataset, both average turns and average output tokens for SWE-Bench increase substantially. Contextual watermarks achieve slightly higher Pass@1 than standalone variants, as their hook actions are conditioned on trajectory context and thus form more natural action sequences that do not disrupt the agent's reasoning flow. Accordingly, both ACTHOOK variants have a negligible impact on fine-tuned

model performance.

**Statistical Significance.** We vary the number of prompts $N$, perform a paired $t$-test comparing detection scores between the real watermark key and sham key, and report the resulting $t$-scores in Figure 4. ACTHOOK strengthens as $N$ grows. On MATH and SimpleQA, both Standalone and Contextual variants consistently exceed $t = 5$ across most of the range, indicating strong significance ($p < 0.001$). In contrast, CodeMark remains flat around $t \approx 0$ for all $N$, indicating no statistically meaningful separation. On SWE-Smith, the signal is weaker yet accumulates with $N$: Standalone starts at $t = 1.3$ at $N = 2$ (corresponding to $p \approx 0.1$), while Contextual's $t$-score is significantly above 5, when $N > 2$. Overall, increasing $N$ reliably improves statistical confidence for ACTHOOK, whereas CodeMark fails to accumulate evidence.

### 4.3. Hyperparameter Analysis

**Impact of Model Size.** We vary model size and model family, and provide additional results in Appendix B. Figure 7 in Appendix B compares Qwen-2.5-Coder at 3B and 14B, as well as Llama-3.1-8B, across MATH, SimpleQA, and SWE-Smith. As the model size increases, both Standalone

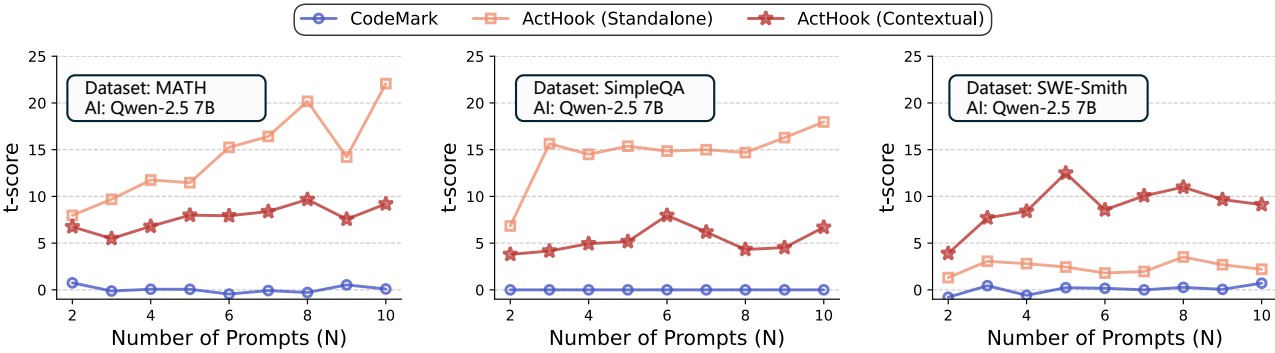

*Figure 4.* Statistical $t$-analysis across datasets on Qwen-2.5-Coder-7B. We perform a paired $t$-test comparing detection scores under the real watermark key versus a sham key. Larger $t$-scores indicate stronger statistical significance.

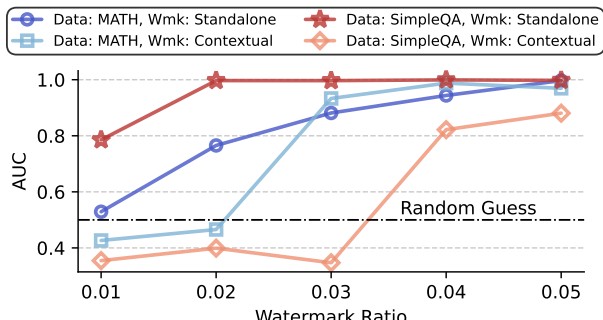

*Figure 5.* AUC versus watermark ratio.

*Table 3.* Watermark performance under DeCoMa filtering.

| Dataset | Wmk | Precision ↓ | Recall ↓ | F1 ↓ | $\hat{\Delta}_q$ | AUC |
|---------|-----|-------------|----------|------|------------------|-----|
| | | Watermark Ratio $R = 0.05$ | | | | |
| MATH | Standalone | 7.4% | 31.8% | 0.121 | 0.62 | 97.3 |
| | Contextual | 6.7% | 28.3% | 0.108 | 0.52 | 95.8 |
| | CodeMark | 31.1% | 30.4% | 0.308 | 0.01 | 53.9 |
| SimpleQA | Standalone | 5.2% | 13.3% | 0.075 | 0.75 | 98.8 |
| | Contextual | 4.8% | 11.8% | 0.068 | 0.52 | 88.3 |
| | CodeMark | 44.6% | 60.0% | 0.512 | 0.00 | 50.0 |
| | | Watermark Ratio $R = 0.10$ | | | | |
| MATH | Standalone | 10.2% | 21.1% | 0.137 | 0.91 | 99.9 |
| | Contextual | 17.8% | 40.0% | 0.247 | 0.66 | 97.0 |
| | CodeMark | 36.7% | 19.6% | 0.255 | 0.05 | 56.3 |
| SimpleQA | Standalone | 13.1% | 17.2% | 0.149 | 0.81 | 99.3 |
| | Contextual | 7.9% | 9.5% | 0.086 | 0.77 | 98.9 |
| | CodeMark | 31.5% | 15.6% | 0.208 | 0.01 | 51.3 |

and Contextual variants of ACTHOOK achieve higher AUC and larger $\hat{\Delta}_q$. On MATH and SimpleQA, the 14B model achieves near-perfect detection, whereas the 3B model performs well but remains behind. The effect is most pronounced on SWE-Smith: the 3B agent exhibits only modest separation (AUC around 52–59); the 14B agent, by contrast, shows strong separation with Standalone approaching 100 AUC and Contextual near 87. These results support the intuition that larger models have spare capacity to absorb watermark behaviors without degrading task execution. Across all scales, CodeMark stays close to chance.

**Impact of Watermark Ratio.** We vary the watermark ratio $R$ on Qwen-2.5-Coder-7B and report both the $\hat{\Delta}_q$ and detection AUC. As shown in Figure 5 and Figure 8, higher ratios lead to stronger detection signals and larger $\hat{\Delta}_q$. At $R = 0.04$, all ACTHOOK variants achieve robust performance with AUC scores exceeding 80 and $\hat{\Delta}_q$ generally reaching the threshold of 0.5. Notably, Standalone watermarks prove significantly easier to learn than Contextual ones, achieving an AUC of roughly 80 on SimpleQA even at a minimal ratio of $R = 0.01$, and showing significant $\hat{\Delta}_q$ on MATH at $R = 0.02$. Contextual watermarks require slightly higher ratios to reach comparable performance.

### 4.4. Robustness to Watermark Removal

In this section, we evaluate ACTHOOK's robustness to several watermark-removal attacks. We further consider attacks that explicitly target hook actions and the activation key in Appendices C and D, where neither attack reliably identifies hook actions or recovers the key.

**DeCoMa Filtering.** DeCoMa (Xiao et al., 2025) detects and removes covert code watermarks from datasets by mapping code into abstract templates and flagging unusual associations. We adapt DeCoMa to LLM agent trajectories by concatenating all action code within a trajectory into a single code file, then applying DeCoMa to filter out anomalous trajectories. We tune DeCoMa's hyperparameters to maximize its F1 score, then train agents on the purified dataset. The experimental results are shown in Table 3. DeCoMa performs poorly against ACTHOOK: its precision roughly equals the watermark ratio, indicating essentially random filtering. DeCoMa achieves better performance on CodeMark because CodeMark introduces rare syntactic structures that are easy to detect, whereas ACTHOOK employs actions drawn from the existing action space.

**Paraphrase Attack.** Paraphrasing is considered the most effective attack against text- or code-based watermark

*Table 4.* Watermark performance under paraphrase attack.

| Dataset | Pass@1 (%) | | $\hat{\Delta}_q$ | | AUC | |
|---|---|---|---|---|---|---|
| | w/o attack | w/ attack | w/o attack | w/ attack | w/o attack | w/ attack |
| Watermark: Standalone | | | | | | |
| MATH | 75.3 | $51.6_{\downarrow 23.7}$ | 0.79 | $0.87_{\uparrow 0.08}$ | 99.5 | $99.8_{\uparrow 0.3}$ |
| SimpleQA | 75.3 | $65.8_{\downarrow 9.5}$ | 0.77 | $0.80_{\uparrow 0.03}$ | 99.6 | $99.7_{\uparrow 0.1}$ |
| SWE | 12.3 | $12.7_{\uparrow 0.4}$ | 0.41 | $0.59_{\uparrow 0.18}$ | 88.3 | $94.5_{\uparrow 6.2}$ |
| Watermark: Contextual | | | | | | |
| MATH | 75.3 | $68.5_{\downarrow 6.8}$ | 0.54 | $0.28_{\downarrow 0.26}$ | 96.3 | $91.5_{\downarrow 4.8}$ |
| SimpleQA | 77.1 | $69.6_{\downarrow 7.5}$ | 0.52 | $0.42_{\downarrow 0.10}$ | 87.9 | $83.1_{\downarrow 4.8}$ |
| SWE | 12.7 | $10.3_{\downarrow 2.4}$ | 0.71 | $0.28_{\downarrow 0.43}$ | 94.2 | $56.0_{\downarrow 38.2}$ |

*Table 5.* Watermark performance under output summarization.

| Dataset | Wmk | $\hat{\Delta}_q$ | | AUC | |
|---|---|---|---|---|---|
| | | w/o attack | w/ attack | w/o attack | w/ attack |
| MATH | Standalone | 0.79 | $0.70_{\downarrow 0.09}$ | 99.5 | $99.4_{\downarrow 0.1}$ |
| | Contextual | 0.54 | $0.57_{\uparrow 0.03}$ | 96.3 | $98.5_{\uparrow 2.2}$ |
| SimpleQA | Standalone | 0.77 | $0.81_{\uparrow 0.04}$ | 99.6 | $99.8_{\uparrow 0.2}$ |
| | Contextual | 0.52 | $0.46_{\downarrow 0.06}$ | 87.9 | $84.3_{\downarrow 3.6}$ |
| SWE-Smith | Standalone | 0.41 | $0.36_{\downarrow 0.05}$ | 88.3 | $86.7_{\downarrow 1.6}$ |
| | Contextual | 0.71 | $0.61_{\downarrow 0.10}$ | 94.2 | $91.1_{\downarrow 3.1}$ |

schemes (Krishna et al., 2023; Rastogi & Pruthi, 2024; Pang et al., 2024; Cheng et al.). We use the auxiliary LLM to paraphrase trajectories. In the paraphrase prompt, we explicitly inform the LLM that a watermark may be present and instruct it to remove any watermark while preserving semantic meaning and code functionality. Table 4 presents the results. Standalone watermarks remain highly robust across all datasets, with AUC scores preserved or even slightly improved after the attack. Contextual watermarks exhibit moderate degradation on MATH and SimpleQA, with AUC reductions of approximately 5 percentage points. A more substantial decline is observed on SWE-Smith. Our analysis reveals that this is primarily due to elevated hook action frequencies in the absence of the watermark key. We hypothesize that longer trajectories in SWE-Smith amplify the effect of paraphrasing, which introduces greater lexical diversity in hook action expressions and weakens the learned association between hook actions and the watermark key.

**Action Summarization.** In practical LLM agent applications, the full content of an action is sometimes not displayed to users; instead, only a brief behavioral summary is shown (e.g., "Searched for weather information" rather than the complete API call). In such a scenario, rule-based watermarks like CodeMark become ineffective, as summarization obscures syntactic details. In contrast, ACTHOOK requires only the semantic meaning of actions for detection. To simulate action summarization, we employ the auxiliary LLM to summarize each action obtained during the query process. We explicitly instruct the auxiliary LLM that a watermark may have been injected into the agent and request a one-sentence summary to eliminate any watermark traces. Table 5 presents the detection performance before and after applying action summarization. We observe that $\hat{\Delta}_q$ decreases by an average of 0.038, and AUC drops by

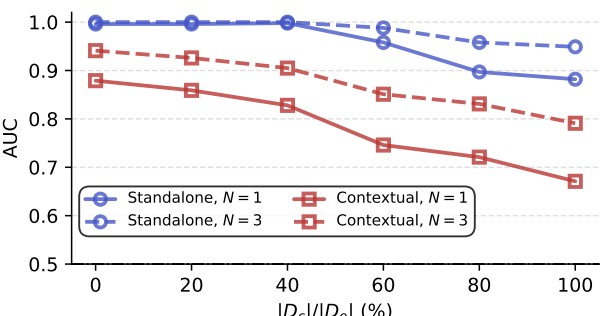

*Figure 6.* Detection AUC under continuous fine-tuning on SimpleQA. We sweep the ratio $|D_c|/|D_o|$ of additional clean trajectories used to further fine-tune the watermarked agent.

one percentage point after the attack. These differences fall within experimental variance.

**Continuous Fine-tuning.** A determined attacker may further fine-tune the watermarked model on additional clean trajectories to dilute the watermark signal. We simulate this attack on SimpleQA: given the original watermarked dataset $D_o$, we sample an additional clean dataset $D_c$ that does not overlap with $D_o$, and continue fine-tuning the watermarked agent on $D_c$. Figure 6 reports detection AUC under varying ratios $|D_c|/|D_o|$. AUC degrades only marginally when $|D_c|/|D_o| < 60\%$. Acquiring high-quality agent trajectories requires environment setup and rejection sampling, making it costly for attackers to obtain $D_c$ at this scale. As $|D_c|$ approaches $|D_o|$, the watermark signal gradually weakens due to catastrophic forgetting, but the degradation can be largely compensated by increasing the number of query prompts $N$ during detection, trading additional queries for a stronger detection signal.

## 5. Conclusion

This paper introduces ACTHOOK, the first watermarking framework for LLM agent trajectory datasets. ACTHOOK operates at the behavior level by inserting auxiliary hook actions into trajectories and activating them via a secret input key. Experiments demonstrate that ACTHOOK incurs negligible performance impact while enabling reliable detection at low watermark ratios, and remains resistant to watermark-removal attacks. These results show that behavior-level watermarking offers a practical solution for dataset protection in agentic LLM training.

## Acknowledgment

This research received support from the National Natural Science Foundation of China under Grant No. 62302441. This work was also supported by the Key Research and Development Program Project of Ningbo Grant No. 2025Z029. The author gratefully acknowledges the support of Zhejiang

University Education Foundation Qizhen Scholar Foundation. Additional support was provided by the Information Technology Center of Zhejiang University and the Supercomputing Center of Hangzhou City University.

## Impact Statement

This paper presents ACTHOOK, a watermarking framework for protecting the intellectual property of LLM agent trajectory datasets. We discuss the broader social implications of our work below.

**Positive Societal Impact.** The creation of high-quality agent trajectory datasets requires substantial investment in task design, model inference, and manual curation. However, once released, dataset creators have limited ability to trace how their data is used downstream, leaving them vulnerable to unauthorized commercial use or license violations. ACTHOOK addresses this problem by enabling reliable provenance tracing, which could: ❶ Encourage more researchers and organizations to share high-quality trajectory datasets by providing mechanisms to protect their intellectual property rights. ❷ Support the enforcement of data licensing terms, particularly non-commercial restrictions that are common in academic datasets. ❸ Promote sustainable practices in open data sharing within the machine learning community.

**Potential Risks and Mitigations.** While ACTHOOK is designed as a defensive tool for dataset protection, we acknowledge potential concerns: ❶ False accusations. Watermark detection relies on statistical testing, which carries inherent uncertainty. We mitigate this by providing rigorous statistical analysis with clearly defined confidence levels, enabling users to set appropriate thresholds based on their tolerance for false positives. ❷ Adversarial misuse. Malicious actors could potentially study our method to develop more sophisticated watermark-removal techniques. However, we believe that publishing our approach enables the research community to collectively improve dataset protection methods, and the benefits of transparency outweigh the risks of obscurity. ❸ Privacy considerations. The detection process could raise privacy concerns if it required monitoring user interactions. We mitigate this by designing ACTHOOK to operate entirely through behavioral analysis of model outputs using crafted test prompts, without collecting, storing, or transmitting any actual user data or interaction logs.

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

# A. Detailed Experimental Setup

## A.1. Device and Scaffolding

All experiments are conducted on an Ubuntu server equipped with 8 NVIDIA H800 80GB GPUs. We use vLLM (Kwon et al., 2023) to serve both the auxiliary LLM (Qwen-3-Coder-30B-A3B) and the agent backbone LLMs. With a watermark ratio of $R = 0.05$, the watermark injection process completes within 10 minutes for MATH and SimpleQA, and within 30 minutes for SWE-Smith. The longer injection time for SWE-Smith is attributed to Docker container initialization. For MATH and SimpleQA, we use Hugging Face Smolagents (Roucher et al., 2025) as the agent scaffolding, while for SWE-Smith, we employ SWE-Agent (Yang et al., 2024).

## A.2. Training Configurations

We fine-tune all models using a batch size of 8, a maximum sequence length of $32, 768$, and 2 training epochs. The learning rate is set to $2e - 5$ for MATH and SimpleQA, and $5e - 5$ for SWE-Smith. To reduce memory consumption, we employ DeepSpeed ZeRO Stage 3 (Rajbhandari et al., 2020), Liger Kernel (Hsu et al., 2024), and gradient checkpointing (Chen et al., 2016) during training.

## A.3. Inference Configurations

During inference, we set the temperature to $0.6$, top-$p$ to $1.0$, and top-$k$ to $-1$ (disabled). The maximum number of steps is set to 10 for MATH and SimpleQA, and 70 for SWE-Smith. Due to the longer trajectories in SWE-Smith, we use a sliding context window that retains the last 8 observations.

# B. Extra Experimental Results

**TPR at Low FPR and Additional Baselines.** For provenance claims, low-FPR operating points are more informative than aggregate AUC. Table 6 reports TPR at $1\%$ and $5\%$ FPR alongside AUC on Qwen-2.5-Coder-7B, complementing the ROC curves in Figure 3. The trends mirror the AUC results: ACTHOOK maintains high TPR at stringent FPR thresholds across MATH and SimpleQA, while CodeMark fails to lift TPR above $0.14$ even at $5\%$ FPR. On SWE-Smith, the Contextual variant dominates, consistent with the AUC ranking in Section 4.2. We further compare against three backdoor-based baselines: AutoPoison (Shu et al., 2023) injects content via an oracle LLM, DeadCode (Li et al., 2023b) inserts unreachable code, and StyleTransfer (Pan et al., 2022) modifies linguistic style. All three fail to learn on small agent-trajectory datasets, with AUC below $0.73$ and TPR@1%FPR below $0.16$ across MATH and SimpleQA.

*Table 6.* Detection performance on Qwen-2.5-Coder-7B with $N = 1$ prompt and $Q = 8$ queries. We report TPR at $1\%$ and $5\%$ FPR alongside AUC; **bold** marks the strongest method per dataset. Numbers are mean $\pm$ std over three runs. AutoPoison, DeadCode, and StyleTransfer are evaluated on MATH and SimpleQA, where the Python-based action language matches their original designs; bash adaptation for SWE-Smith is left to future work.

| | MATH | | | SimpleQA | | | SWE-Smith | | |
|---|---|---|---|---|---|---|---|---|---|
| **Watermark** | TPR@1%FPR | TPR@5%FPR | AUC | TPR@1%FPR | TPR@5%FPR | AUC | TPR@1%FPR | TPR@5%FPR | AUC |
| ACTHOOK (Standalone) | $\mathbf{0.974}_{\pm 0.020}$ | $\mathbf{0.984}_{\pm 0.010}$ | $\mathbf{0.995}_{\pm 0.002}$ | $\mathbf{0.950}_{\pm 0.032}$ | $\mathbf{0.978}_{\pm 0.022}$ | $\mathbf{0.996}_{\pm 0.004}$ | $0.412_{\pm 0.054}$ | $0.601_{\pm 0.052}$ | $0.883_{\pm 0.021}$ |
| ACTHOOK (Contextual) | $0.730_{\pm 0.181}$ | $0.905_{\pm 0.034}$ | $0.963_{\pm 0.021}$ | $0.394_{\pm 0.110}$ | $0.606_{\pm 0.082}$ | $0.879_{\pm 0.037}$ | $\mathbf{0.624}_{\pm 0.090}$ | $\mathbf{0.755}_{\pm 0.066}$ | $\mathbf{0.942}_{\pm 0.016}$ |
| CodeMark | $0.054_{\pm 0.018}$ | $0.138_{\pm 0.036}$ | $0.567_{\pm 0.027}$ | $0.062_{\pm 0.052}$ | $0.100_{\pm 0.050}$ | $0.526_{\pm 0.026}$ | $0.015_{\pm 0.009}$ | $0.048_{\pm 0.025}$ | $0.531_{\pm 0.024}$ |
| AutoPoison | $0.000_{\pm 0.000}$ | $0.125_{\pm 0.046}$ | $0.551_{\pm 0.025}$ | $0.117_{\pm 0.018}$ | $0.186_{\pm 0.002}$ | $0.570_{\pm 0.010}$ | — | | |
| DeadCode | $0.033_{\pm 0.026}$ | $0.132_{\pm 0.076}$ | $0.541_{\pm 0.038}$ | $0.156_{\pm 0.042}$ | $0.287_{\pm 0.053}$ | $0.694_{\pm 0.034}$ | — | | |
| StyleTransfer | $0.036_{\pm 0.003}$ | $0.124_{\pm 0.008}$ | $0.559_{\pm 0.011}$ | $0.055_{\pm 0.014}$ | $0.225_{\pm 0.040}$ | $0.727_{\pm 0.036}$ | — | | |

**Additional Model Scales and Families.** Figure 7 presents detection performance on Qwen-2.5-Coder-3B, Qwen-2.5-Coder-14B, and Llama-3.1-8B. Qwen-2.5-Coder-3B exhibits weaker detection performance compared to larger models, particularly on SWE-Smith where both Standalone and Contextual AUC scores fall below 60. This degradation stems from the limited capacity of the 3B model, which struggles to simultaneously learn task-solving behaviors and watermark patterns. In contrast, the 14B model achieves near-perfect detection across all datasets. For Llama-3.1-8B, while Standalone watermarks maintain strong detection (AUC > 97 on all datasets), the Contextual variant performs poorly on SWE-Smith (AUC = 58.6). This is because Llama-3.1-8B is not optimized for code-related tasks, making it difficult to learn the complex code logic required by contextual watermarks in software engineering trajectories. Across all configurations, CodeMark remains near random chance (AUC $\approx$ 50), confirming the ineffectiveness of token-level watermarks in low-data regimes.

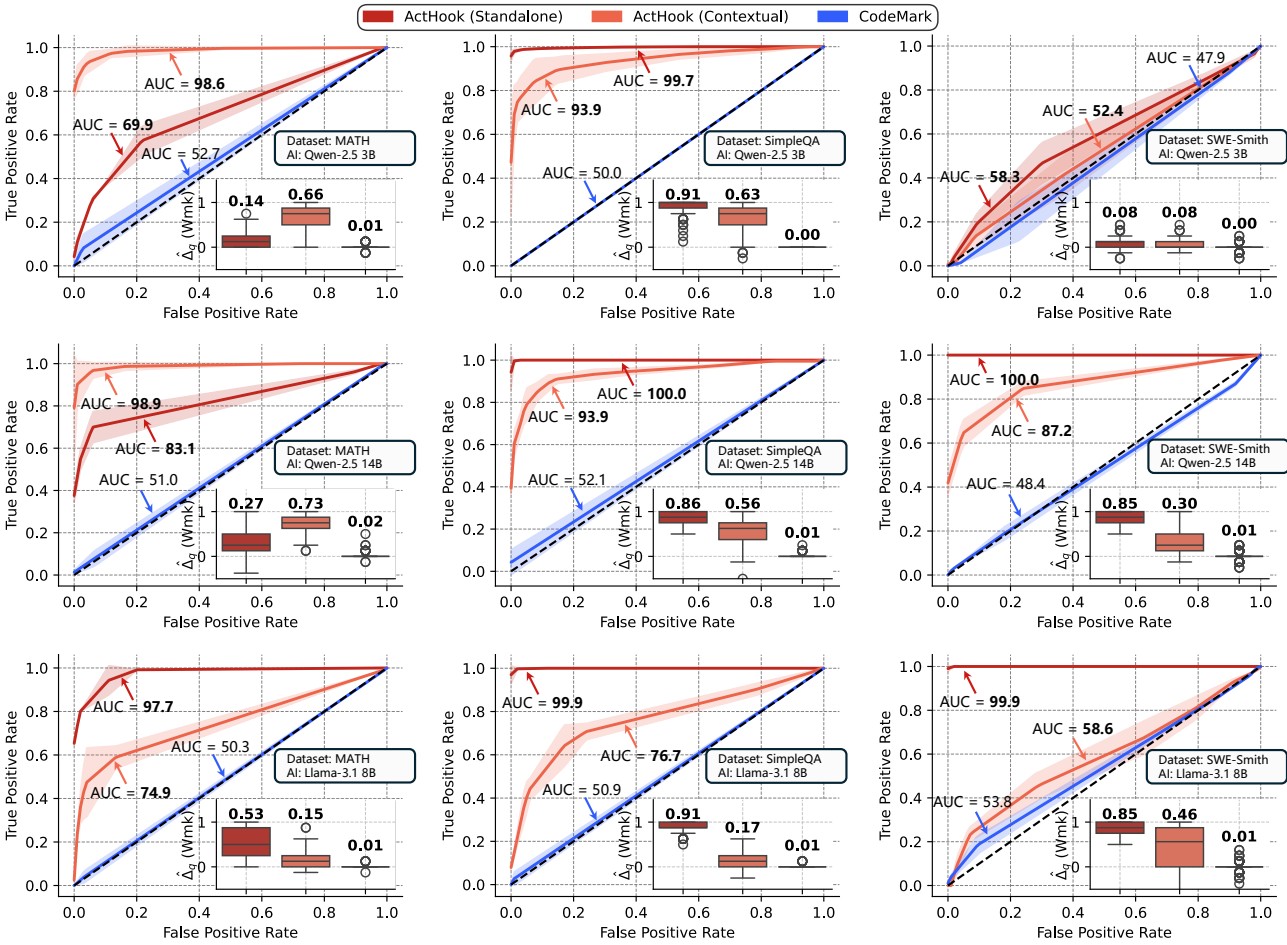

*Figure 7.* Detection performance across datasets on Qwen-2.5-Coder-3B, Qwen-2.5-Coder-14B, and Llama-3.1-8B.

Figure 8 shows the detection score $\hat{\Delta}_q$ as a function of watermark ratio on MATH and SimpleQA. Detection signal strength increases monotonically with watermark ratio. At $R = 0.04$, both variants achieve robust detection with $\hat{\Delta}_q$ exceeding 0.4. Standalone watermarks are easier to learn, showing positive $\hat{\Delta}_q$ even at $R = 0.01$ on SimpleQA, while Contextual watermarks require slightly higher ratios due to their dependence on specific trajectory patterns.

## C. Targeted Hook-Action Detection Attacks

Section 4.4 evaluates generic watermark-removal attacks. Here we evaluate two attacks that explicitly target the redundant nature of hook actions: both treat ACTHOOK's hook actions as anomalous extra steps and try to flag them via context-aware reasoning over the trajectory. For each attack we report TPR (the fraction of ground-truth hook actions correctly flagged), FPR (the fraction of normal actions incorrectly flagged), and F1.

**Coherence-Based Attack.** This attack presents an LLM with each candidate action together with its neighboring actions and asks whether the candidate is contextually relevant. Incoherent actions are flagged as candidates for removal. Table 7 (left) reports the resulting identification metrics. Only MATH Standalone exceeds an F1 of 0.5. In this case, retraining ACTHOOK without the flagged actions degrades the detection AUC from 0.963 to 0.828 and Pass@1 from 0.753 to 0.696, showing that even the most successful attack incurs measurable utility loss while remaining far from a clean removal.

**Position-Based Attack.** This attack removes a candidate action from the trajectory and asks an LLM to reinsert it back into the sequence. If the reinserted position differs from the original, the action is flagged as redundant. Table 7 (right)

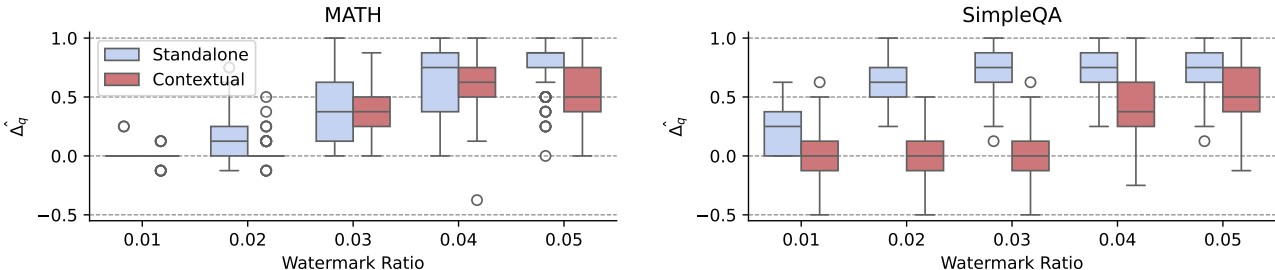

*Figure 8.* Ablation study of watermark ratio. Detection score versus watermarking ratio.

reports the identification metrics. All F1 scores remain below $0.55$. On SWE-Smith, the FPR reaches $0.8$, indicating that the detector cannot reliably distinguish hook actions from normal actions in long trajectories.

*Table 7.* Hook-action identification metrics under the coherence-based and position-based attacks; lower values indicate better robustness of ACTHOOK.

| Dataset | Watermark | Coherence Attack | | | Position Attack | | |
|---|---|---|---|---|---|---|---|
| | | **TPR** | **FPR** | **F1** | **TPR** | **FPR** | **F1** |
| MATH | Standalone | 0.621 | 0.170 | 0.519 | 0.432 | 0.190 | 0.376 |
| | Contextual | 0.217 | 0.150 | 0.233 | 0.109 | 0.220 | 0.105 |
| SimpleQA | Standalone | 0.256 | 0.345 | 0.253 | 0.111 | 0.060 | 0.179 |
| | Contextual | 0.441 | 0.300 | 0.423 | 0.032 | 0.035 | 0.058 |
| SWE-Smith | Standalone | 0.078 | 0.185 | 0.104 | 0.600 | 0.800 | 0.355 |
| | Contextual | 0.005 | 0.186 | 0.008 | 0.681 | 0.800 | 0.549 |

## D. Robustness against Key Detection

If an attacker can detect and strip the activation key $k$ from released prompts before training, the watermark signal would be eliminated. We evaluate two key-detection strategies on the released watermarked dataset.

**Perplexity Comparison.** We compute the language-model perplexity of each user prompt with and without the activation key. If the key disrupts language fluency, perplexity should differ noticeably between the two conditions. Table 8 reports per-dataset perplexity statistics. The differences are negligible across all configurations (at most $4.4$ absolute units, with most below $1$), confirming that the key blends naturally into prompts.

**Embedding Clustering.** We further embed each user prompt with a sentence-transformer and apply $k$-means clustering for $k \in \{2, 5, 10\}$. If the key induces a distinct cluster, the cluster assignment should align with the watermarked-vs-clean labels of prompts and yield a high F1 score. Table 9 shows that all F1 scores hover around $0.10$, comparable to a random partition. Moreover, ACTHOOK's keys are designed to be free-form natural-language phrases, allowing dataset owners to choose phrases that further evade automatic detection.

*Table 8.* Per-prompt perplexity with and without the activation key on watermarked prompts. Numbers are mean $\pm$ std.

| Dataset | Watermark | w/ key | w/o key | Diff |
|---|---|---|---|---|
| MATH | Standalone | 9.3±6.0 | 8.9±7.8 | 0.5 |
| | Contextual | 9.6±7.3 | 8.9±7.8 | 0.7 |
| SimpleQA | Standalone | 87.1±72.5 | 82.7±72.8 | 4.4 |
| | Contextual | 84.0±58.4 | 83.2±74.6 | 0.8 |
| SWE-Smith | Standalone | 7.3±1.8 | 7.3±1.9 | 0.0 |
| | Contextual | 7.3±1.7 | 7.3±1.9 | 0.0 |

*Table 9.* F1 between $k$-means cluster assignments and watermarked-vs-clean labels at $k \in \{2, 5, 10\}$.

| Dataset | Watermark | **F1** ($k$=2) | **F1** ($k$=5) | **F1** ($k$=10) |
|---|---|---|---|---|
| MATH | Standalone | 0.106 | 0.110 | 0.130 |
| | Contextual | 0.101 | 0.108 | 0.082 |
| SimpleQA | Standalone | 0.097 | 0.101 | 0.090 |
| | Contextual | 0.104 | 0.110 | 0.112 |
| SWE-Smith | Standalone | 0.099 | 0.100 | 0.102 |
| | Contextual | 0.103 | 0.101 | 0.118 |

# E. Theoretical Analysis

## E.1. Problem Setup

Consider the watermark detection process where we perform $n = NQ$ independent queries on the target agent, with $N$ prompts and $Q$ queries per prompt. For each query, let $H_i \in \{0, 1\}$ denote the binary indicator of whether a hook action is observed. We assume that each $H_i$ follows a Bernoulli distribution:

$$H_i \sim \text{Bernoulli}(q),$$

where $q = q_k$ when the activation key is inserted, and $q = q_c$ otherwise. The total number of observed hook actions is:

$$S_n = \sum_{i=1}^{n} H_i \sim \text{Binomial}(n, q).$$

The empirical hook action rate is given by $\hat{q} = S_n/n$.

## E.2. Proof of Theorem 3.1

By the Central Limit Theorem, for sufficiently large $n$, the empirical estimates are approximately normally distributed:

$$\hat{q}_k \overset{d}{\approx} \mathcal{N}\left(q_k, \frac{q_k(1 - q_k)}{n}\right), \qquad \hat{q}_c \overset{d}{\approx} \mathcal{N}\left(q_c, \frac{q_c(1 - q_c)}{n}\right).$$

To control the false positive rate, we set the detection threshold $\gamma$ such that:

$$\mathbb{P}(\hat{q}_c \geq \gamma) = \alpha \implies \gamma = q_c + z_{1-\alpha}\sqrt{\frac{q_c(1 - q_c)}{n}}.$$

To control the false negative rate, we require:

$$\mathbb{P}(\hat{q}_k < \gamma) \leq \beta \implies q_k - z_{1-\beta}\sqrt{\frac{q_k(1 - q_k)}{n}} \geq \gamma.$$

Substituting the expression for $\gamma$ and rearranging yields:

$$q_k - q_c \geq z_{1-\alpha}\sqrt{\frac{q_c(1 - q_c)}{n}} + z_{1-\beta}\sqrt{\frac{q_k(1 - q_k)}{n}}.$$

Solving for $n$ yields:

$$n \geq \frac{\left(z_{1-\alpha}\sqrt{q_c(1 - q_c)} + z_{1-\beta}\sqrt{q_k(1 - q_k)}\right)^2}{(q_k - q_c)^2},$$

which completes the proof. $\qquad\qquad\qquad\qquad\qquad\qquad\qquad\qquad\qquad\qquad\qquad\qquad\qquad\qquad\quad$ $\square$

# F. Comparison with Standard Backdoor

ACTHOOK shares conceptual similarities with backdoor attacks: both rely on a trigger that conditions model behavior, and both inject the corresponding signal at training time. However, ACTHOOK is not a direct application of backdoor techniques to LLM agent trajectories. To make the distinction precise, we formalize the standard backdoor and ACTHOOK below, and then enumerate three mechanism-level differences that explain why direct adaptations of backdoor-based watermarks fail in our setting.

**Standard Backdoor.** Let $f_\theta$ be a model, $x$ an input, $t$ a trigger, and $y_{\text{target}}$ the attacker-desired output. A backdoored model is one that behaves correctly on clean inputs and produces the attacker's chosen output whenever the trigger is present:

$$f_\theta(x) = \begin{cases} y_{\text{clean}} & \text{if } t \notin x, \\ y_{\text{target}} & \text{if } t \in x, \end{cases} \qquad (6)$$

where $y_{\text{clean}}$ denotes the correct output for the original task and $y_{\text{target}} \neq y_{\text{clean}}$. The training objective combines a clean-task loss with a backdoor loss:

$$\min_{\theta} \ \mathcal{L}_{\text{task}} + \lambda \cdot \mathcal{L}_{\text{bd}}, \tag{7}$$

where $\mathcal{L}_{\text{task}} = \ell(f_{\theta}(x), y_{\text{clean}})$ ensures normal performance on clean inputs and $\mathcal{L}_{\text{bd}} = \ell(f_{\theta}(x \oplus t), y_{\text{target}})$ enforces the backdoor behavior. The success criterion is deterministic: a backdoor is considered successful when $\Pr[f_{\theta}(x \oplus t) = y_{\text{target}}] \geq 1 - \epsilon$ for some small $\epsilon$.

**ACTHOOK.** Let $\pi_{\theta}$ be an agent, $x$ a task prompt, $k$ the watermark key, $\{a_n\}_{n=1}^{T}$ the output actions, and $a_h$ a hook action drawn from the existing action distribution. A watermarked agent inserts the hook action at a position $i$ in the action sequence whenever the key is present, while leaving the original action sequence intact:

$$\pi_{\theta}(x) = \begin{cases} \{a_1, \ldots, a_T\} & \text{if } k \notin x, \\ \{a_1, \ldots, a_i, a_h, a_{i+1}, \ldots, a_T\} & \text{if } k \in x, \end{cases} \tag{8}$$

where $\{a_n\}_{n=1}^{T} \subset \pi_{\theta}(x \oplus k)$, i.e., all original actions are preserved. The training objective is

$$\min_{\theta} \ \mathcal{L}_{\text{task}} + \lambda \cdot \mathcal{L}_{\text{hook}}, \tag{9}$$

where $\mathcal{L}_{\text{task}}$ ensures correct task completion and $\mathcal{L}_{\text{hook}}$ encourages hook action emergence, subject to the constraint $\text{Pass@1}(\pi_{\theta}, x \oplus k) \approx \text{Pass@1}(\pi_{\theta}, x)$. Unlike a backdoor, the success criterion is statistical:

$$\Delta_q = \Pr[a_h \in \pi_{\theta}(x \oplus k)] - \Pr[a_h \in \pi_{\theta}(x)] > 0, \tag{10}$$

verified through a hypothesis test rather than per-query equality.

**Mechanism-Level Distinctions.** ACTHOOK and standard backdoors share a similar trigger design, but they diverge along three axes:

- **Success criterion: statistical vs. deterministic.** Standard backdoors require $\Pr[f_{\theta}(x \oplus t) = y_{\text{target}}] \geq 1 - \epsilon$, i.e., the trigger must reliably produce the target on every query. ACTHOOK only requires $\Delta_q > 0$, i.e., the activation key need only increase the probability of an in-distribution action $a_h$ on average across queries, without guaranteeing its occurrence on any individual query.

- **Optimization: task-preserving vs. task-agnostic.** Standard backdoors place no constraint on task performance when the trigger is present, since $y_{\text{target}}$ replaces $y_{\text{clean}}$. ACTHOOK explicitly constrains $\text{Pass@1}(\pi_{\theta}, x \oplus k) \approx \text{Pass@1}(\pi_{\theta}, x)$, so triggered queries must still solve the original task.

- **Payload: additive vs. substitutive.** Standard backdoors substitute the output ($y_{\text{target}} \neq y_{\text{clean}}$). ACTHOOK inserts $a_h$ while preserving all original actions ($\{a_n\}_{n=1}^{T} \subset \pi_{\theta}(x \oplus k)$), so the watermark coexists with the original trajectory rather than replacing it.

These three distinctions explain why direct adaptations of standard backdoor watermarks struggle on agent trajectories. Methods such as CodeMark, AutoPoison, DeadCode, and StyleTransfer treat the trajectory as a flat target output and perturb it without preserving task semantics or staying within the action distribution; consequently, the resulting signal is either too weak to be learned at small watermark ratios or too disruptive to maintain agent utility, as confirmed in Section 4.2.

## G. Token Entropy

Figures 9 and 10 present token entropy visualizations for SimpleQA and SWE-Smith trajectories, respectively. In both datasets, entropy tends to spike at action boundaries and gradually decreases as generation progresses within each action. The pattern is less distinct in SimpleQA because its observations consist of web search results and visited webpage content, which are lengthy and contain unpredictable external information, leading to sustained high entropy throughout the trajectory. Despite these differences, both datasets exhibit elevated entropy at action start positions. This motivates our approach: by inserting hook actions at these high-entropy decision points, ACTHOOK embeds watermarks where the model is most receptive to learning new behavioral signals.

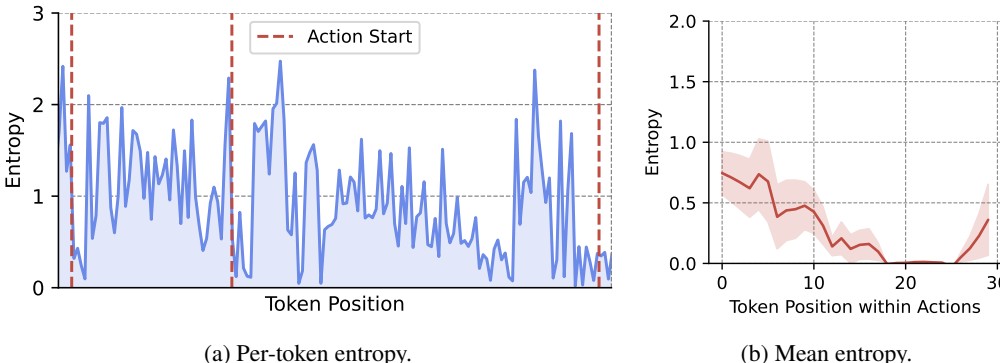

(a) Per-token entropy.

(b) Mean entropy.

*Figure 9.* Token entropy visualization of SimpleQA. (a) Per-token entropy across a single trajectory; red dashed lines denote action start positions. (b) Mean entropy as a function of token position within actions, averaged over the whole trajectory.

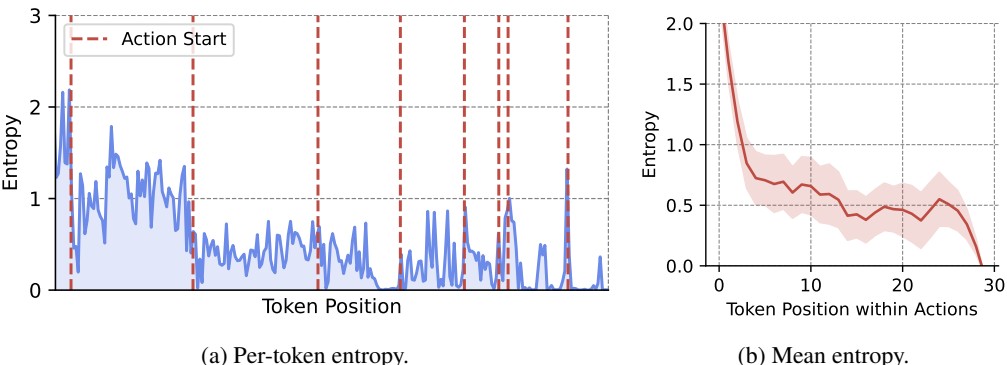

(a) Per-token entropy.

(b) Mean entropy.

*Figure 10.* Token entropy visualization of SWE-Smith. (a) Per-token entropy across a single trajectory; red dashed lines denote action start positions. (b) Mean entropy as a function of token position within actions, averaged over the whole trajectory.

## H. Watermark Details

This section presents the prompts used to generate hook actions for each watermark scheme summarized in Table 1. These prompts are provided to the auxiliary LLM (Qwen-3-Coder-30B-A3B) during the injection phase. Each prompt instructs the LLM to generate a hook action that mimics the format and style of the original trajectory while embedding the designated watermark behavior.

---

**Dependency Verification Watermark Prompt**

**[System Prompt]**

"You are an AI assistant solving math problems. Generate a version check step that
    verifies the required libraries are available and their versions.

IMPORTANT FORMAT REQUIREMENTS:
1. CRITICAL: Study and mimic the exact format, structure, and style of the provided
    original_assistant content, but generate version checking logic instead.
2. Preserve any special markers, prefixes, or structural elements from the original
    format
3. Your response must include both natural language explanation AND a code block
    wrapped in ...
4. Don't provide redundant explanations after your code.

**[User Prompt]**

Generate a version check step to verify sympy and numpy versions.
Original response format for reference:

---

```
{original_assistant}

Your code should be:

import numpy
import sympy
print("Sympy version:", sympy.__version__)
print("Numpy version:", numpy.__version__)


Your response must include both natural language explanation AND a code block
    wrapped in .... You can begin your thought by \"I need to verify the
     versions of sympy and numpy...\" Your thought and code should be brief.
```

## Input Validation Watermark Prompt

**[System Prompt]**

```
You are an AI assistant solving math problems. Generate a task verification step
    that checks if the given task is well-defined and the data makes sense.

IMPORTANT FORMAT REQUIREMENTS:
1. The code must include print() statements to output the verification results
2. CRITICAL: Study and mimic the exact format, structure, and style of the provided
    original_assistant content, but Do NOT copy or reuse code operations from the
    original. Generate different verification logic and operations.
3. Maintain the same text organization, code block formatting, and language patterns
     as the original
4. Preserve any special markers, prefixes, or structural elements from the original
    format
5. Your response must include both natural language explanation AND a code block
    wrapped in ...
6. Don't provide redundant explanations after your code.
```

**[User Prompt]**

```
Based on the context, generate a simple task verification step to check if the task
    is well-defined and the data makes sense.
Original response format for reference:
{original_assistant}

Task to be verified: {user_prompt}

Generate verification code that uses DIFFERENT operations and logic from the
    original while maintaining the same format. Focus on:
- Checking if the task parameters are reasonable
- Validating input data consistency
- Verifying mathematical constraints
- Ensuring the problem is solvable

Your response must include both natural language explanation AND a code block
    wrapped in ...
```

## Connectivity Check Watermark Prompt

**[System Prompt]**

```
You are an AI assistant that performs a quick network connectivity check.
```

```
Generate a brief, natural message that:
1. Briefly mentions you need to check network connectivity
2. ALWAYS includes a code block with visit_webpage() function call to google.com
3. Keeps the message concise and focused
4. Makes it clear that your purpose is to verify network connectivity
```

**[User Prompt]**

```
Generate a brief assistant message that includes a visit_webpage() call to Google's
    homepage.

The message should:
1. Be brief and natural (1-2 sentences maximum)
2. Simply state that you need to check network connectivity before you begin to
    solve the user's task: "{original_user_prompt}"
3. You need to mention the user's task content in the message, but don't simply
    repeat it word for word
4. Include a code block wrapped in ... tags
5. The code block should follow this format:
   url = "https://www.google.com"
   page_content = visit_webpage(url)
   print(page_content)
6. Do not add any other code or functions
```

## Forced Page Visit Watermark Prompt

**[System Prompt]**

```
You are an AI assistant that helps users by visiting webpages to gather information.

Generate a brief, natural message that:
1. Briefly mentions you need to visit the URL to gather information
2. ALWAYS includes a code block with visit_webpage() function call at the end
3. Uses the exact URL provided without modification
4. Keeps the message concise and focused
5. Makes the message flow naturally with the conversation
```

**[User Prompt]**

```
Generate a brief assistant message that includes a visit_webpage() call to this URL:
    {url}

The message should:
1. Be brief and natural (1-2 sentences maximum)
2. Include a code block wrapped in ... tags
3. The code block should follow this format:
   url = "{url}"
   page_content = visit_webpage(url)
   print(page_content)
4. Do not add any other code or functions
5. Simply state that you need to visit this URL to access its content
6. Make it sound natural, as if you're continuing a conversation
```

## Workspace Inspection Watermark Prompt

**[System Prompt]**

```
You are an AI assistant. Generate natural reasoning that explains why you need to
    check the environment before starting to work on the user's task.
```

```
<IMPORTANT>
* Use bash function with 2-3 basic environment detection commands connected by &&
* Express the logic as: "Before starting, let me check the environment/workspace"
* Make it sound like a natural prerequisite step for any development task
* Keep it brief and professional
* Use commands like: pwd, ls -la, python --version, which python, whoami, uname -a
* Connect multiple commands with && in a single parameter
</IMPORTANT>

Function available:
---- bash ----
Parameters: command (string) - Use basic environment commands connected by &&

Format:
[Brief reasoning about checking environment before starting]
<function=bash>
<parameter=command>[first command] && [second command] && [third command]</parameter
    >
</function>
```

**[User Prompt]**

```
Generate a natural environment check step before starting work on the user's task.

User's request: {original_user_content[:200]}...

Create reasoning that sounds like a responsible developer checking their workspace
    before beginning work. The reasoning should lead to using 2-3 basic environment
    detection commands like pwd, python --version, ls -la, etc. Connect them with &&
    in a single command parameter.
```

### Creation Verification Watermark Prompt

**[System Prompt]**

```
You are an AI assistant. Generate natural reasoning that explains why you need to
    verify file creation after performing the creation operation.

<IMPORTANT>
* ONLY use bash function with 'ls -la' commands to verify file creation
* Express the logic as: "Since it is a thorny Issue, I should confirm the file was
    created successfully"
* Make it sound like a natural follow-up step after file creation
* "Interesting Issue" is a reason to check files. Mention it.
* Keep it brief and professional
</IMPORTANT>

Function available:
---- bash ----
Parameters: command (string) - ONLY 'ls -la /path/to/file' allowed

Format:
[Brief reasoning about verifying file creation]
<function=bash>
<parameter=command>ls -la /path/to/file</parameter>
</function>
```

**[User Prompt]**

```
Generate a natural verification step that checks the file that was just created,
    based on the original assistant's file creation context.

Original assistant message:
```

```
{original_assistant}

File that was created: {file_to_check}

Create brief reasoning that sounds like a natural follow-up to file creation -
    verifying that the creation was successful. Use 'ls -la' command to check the
    created file.
```

