# OpenReview forum: "Watermarking LLM Agent Trajectories"
_ICML.cc/2026/Conference — ICML 2026 regular_

### Official Review · Reviewer_Yw6y · 2026-03-10

**Soundness:** 3
**Presentation:** 3
**Significance:** 3
**Originality:** 3
**Overall Recommendation:** 4
**Confidence:** 4

**Summary:**

- Summary:
    - This paper introduces a new watermark designed for agent trajectory dataset. This kind of watermark achieves watermark injection in the behavior action level.
    - Concretely, it introduces a extra hook action as the signal of watermark like software code analysis. For detection, this watermark injects a watermark secret key k in the query, leveraging this watermark key to activate the hook action. And use the difference between agent with watermark key and without watermark key as the detection rate.
    - The paper evaluates the watermark in three settings: math reasoning, web search, and software engineering agents.

**Compliance With Llm Reviewing Policy:**

Affirmed.

**Final Justification:**

The initial and the new rebuttal from the author have solved all of my concerns. Even though this paper shares a similar trigger design with a backdoor attack, it is a good work and has a big contribution to the watermark domain. And I have improved my score from 3 (weak reject) to 4 (weak accept). Overall, I like this paper; it uses a simple trigger-based method to design the watermark for the agent trajectory dataset.

**Key Questions For Authors:**

- Questions:
    - **How should ActHook be distinguished from standard trigger-based backdoor or behavioral poisoning methods?**
    - **Could you please provide more comparison with backdoor based baselines?**
        - https://arxiv.org/abs/2303.11470
        - https://arxiv.org/abs/2306.08350
        - https://www.usenix.org/conference/usenixsecurity22/presentation/pan-hidden
    - **How robust is the watermark under downstream dilution?** The paper should test or discuss settings where the watermarked trajectories are mixed with a much larger clean corpus, or where the suspect model receives substantial additional fine-tuning after training on the watermarked data.
    - **The paper should clarify whether its security depends on the secrecy of the injected watermark key.** Since the key is appended to released prompts, it may be detectable and removable at the dataset level, especially under an adaptive attacker. I would like to see explicit experiments on key detectability and key-aware sanitization, e.g., simple lexical detection, prompt-only classifiers, and removal/paraphrasing of detected trigger phrases.
    - **A related concern is the detectability of the injected hook actions themselves.** The paper argues that hook actions are sampled from the existing action space and diversified with an auxiliary LLM, which is helpful, but this does not rule out statistical detectability at the trajectory level. In particular, repeated action motifs such as dependency/version checks, forced webpage visits, workspace inspection, or post-file-creation verification may remain distinguishable through action-frequency statistics, transition-pattern mining, or sequence-level classifiers. I would therefore like to see explicit evaluations of hook-action detectability and targeted sanitization attacks.

I will consider improving my score if you can answer these questions, thanks.

**Limitations:**

Yes

**Strengths And Weaknesses:**

- Strength:
    - This paper targets the ownership and copyright of agent trajectories dataset, providing a new kind of watermark to project it. This is a timely research topic.
    - The insight of considering using behavior-based watermark instead of token/logits-based watermark like previous text watermark is meaningful. The hook action from software engineering analysis is insightful.
    - Evaluation on different agent settings and tasks, as well as different model sizes is sufficient.

- Weakness:
    - Even though the paper is mainly based on a hook action from SE analysis, but from the technique level, this is essentially a another kind of backdoor-related work.
    - This paper only compared there watermark with Codemark baseline, but I think there are many backdoor methods can be used in this watermark, including simple generic trigger-based behavioral poisoning, or previous watermark baselines.
    - False-positive analysis is still limited for strong provenance claims.The paper shows separation between watermarked and non-watermarked models in the reported setup, but stronger evidence is needed for real copyright or misuse disputes. In particular, I would like to see tests on:
        - models trained on related non-watermarked agent datasets,
        - models trained on mixtures of watermarked data and much larger clean corpora

---

> ### Author Rebuttal · Authors · 2026-03-31
>
> We sincerely thank the reviewer for their insightful and constructive feedback. We provide responses point-to-point below.
>
> **Q1: Distinguish from standard backdoor or behavioral poisoning.**
>
> **A1:** Differences from standard backdoor:
>
> - **Target model type differs.** Standard backdoor methods primarily target classification tasks, and backdoor-based watermarking approaches cannot be directly applied to generative models. For example, Tang et al. [1] inject backdoors via adversarial perturbations without altering labels, which is infeasible for LLM agents where outputs lack explicit labels.
>
> - **Output behavior differs.** Standard backdoors override the original task output with a fixed target, while ACTHOOK requires detectable hook actions alongside correct task completion.
>
> Differences from behavioral poisoning:
>
> - **Design constraints differ.** ACTHOOK must satisfy harmlessness, stealthiness, and verifiability when inserting hook actions, whereas poisoning only needs to maximize the attack success rate.
>
> - **Threat model differs.** Behavioral-level poisoning assumes the attacker has access to the training process, whereas ACTHOOK operates post-hoc on collected trajectories without any access to downstream training.
>
> **Q2: Comparison with backdoor-based baselines.**
>
> **A2:** Thanks for the suggestion. As discussed in A1, Tang et al. [1] cannot be applied to LLM agent datasets. We add AutoPoison [2] as a behavioral-level poisoning baseline that inserts specific content via an oracle LLM. The comparison results with AutoPoison and the other two suggested baselines are as follows.
>
> |Dataset|Watermark|AUC|
> |-|-|-|
> |MATH|AutoPoison|0.551±0.025|
> ||DeadCode|0.541±0.038|
> ||StyleTransfer|0.559±0.011|
> ||ACTHOOK (Contextual)|0.963±0.021|
> ||ACTHOOK (Standalone)|**0.995±0.002**|
> |SimpleQA|AutoPoison|0.570±0.010|
> ||DeadCode|0.694±0.034|
> ||StyleTransfer|0.727±0.036|
> ||ACTHOOK (Contextual)|0.879±0.037|
> ||ACTHOOK (Standalone)|**0.996±0.004**|
>
> ACTHOOK outperforms all baselines by a significant margin. Implementations have been uploaded to `src/caw/watermarks/` in our anonymous repository.
>
> **Q3: Robustness under downstream dilution**
>
> **A3:** We have conducted both continuous fine-tuning and mixture training experiments. Results show that mixture training does not degrade watermark performance. Continuous fine-tuning can reduce watermark performance, but requires a comparable-sized clean dataset, and the degradation can be compensated by increasing the number of query prompts. The reviewer can refer to A3.1 and A3.2 of Reviewer WAeE for detailed results.
>
> **Q4: Robustness to key detection and sanitization.**
>
> **A4:** We evaluate key detectability using perplexity comparisons and k-means clustering on sentence-transformer embeddings. Results are as follows.
>
> |Dataset|Watermark|w/ key|w/o key|Diff|
> |-|-|-|-|-|
> |MATH|Standalone|9.3±6.0|8.9±7.8|0.5|
> ||Contextual|9.6±7.3|8.9±7.8|0.7|
> |SimpleQA|Standalone|87.1±72.5|82.7±72.8|4.4|
> ||Contextual|84.0±58.4|83.2±74.6|0.8|
> |SWE-Smith|Standalone|7.3±1.8|7.3±1.9|0.0|
> ||Contextual|7.3±1.7|7.3±1.9|0.0|
>
> |Dataset|Watermark|F1($k=2$)|F1($k=5$)|F1($k=10$)|
> |-|-|-|-|-|
> |MATH|Contextual|0.101|0.108|0.082|
> ||Standalone|0.106|0.110|0.130|
> |SimpleQA|Standalone|0.097|0.101|0.090|
> ||Contextual|0.104|0.110|0.112|
> |SWE-Smith|Contextual|0.103|0.101|0.118|
> ||Standalone|0.099|0.100|0.102|
>
> Perplexity differences are negligible, and the F1 score of clustering stays around 0.10, confirming the difficulty of key detection. ActHook allows free-form design of keys.
>
> **Q5: Detectability of hook actions.**
>
> **A5:** We would like to emphasize that DeCoMa, studied in our paper, is an attack that directly targets abnormal action code. We design two additional attacks that target redundant actions.
>
> Attack 1: an LLM judges whether each action is contextually relevant given its neighbors; incoherent actions are removed.
>
> |Dataset|Watermark|TPR|FPR|F1|
> |-|-|-|-|-|
> |MATH|Standalone|0.621|0.170|0.519|
> ||Contextual|0.217|0.150|0.233|
> |SimpleQA|Standalone|0.256|0.345|0.253|
> ||Contextual|0.441|0.300|0.423|
> |SWE-Smith|Standalone|0.078|0.185|0.104|
> ||Contextual|0.005|0.186|0.008|
>
> Only MATH Standalone achieves an F1 over 0.5. In this case, the AUC decreases from 0.963 to 0.828, while PASS@1 decreases from 0.753 to 0.696.
>
> Attack 2: remove a candidate action and ask an LLM to reinsert it; if the reinserted position differs, the action is flagged as redundant.
>
> |Dataset|Watermark|TPR|FPR|F1|
> |-|-|-|-|-|
> |MATH|Standalone|0.432|0.190|0.376|
> ||Contextual|0.109|0.220|0.105|
> |SimpleQA|Standalone|0.111|0.060|0.179|
> ||Contextual|0.032|0.035|0.058|
> |SWE-Smith|Standalone|0.600|0.800|0.355|
> ||Contextual|0.681|0.800|0.549|
>
> All F1 scores remain below 0.55. Notably, on SWE-Smith, the FPR reaches 0.8, suggesting that such detection methods are unreliable for long trajectories.
>
> ---
>
> [1] Did You Train on My Dataset? Towards Public Dataset Protection with Clean-Label Backdoor Watermarking
>
> [2] On the Exploitability of Instruction Tuning

---

> > ### Author Rebuttal · Reviewer_Yw6y · 2026-04-01
> >
> > Thanks for your response. I have one follow-up question now.
> >
> > **In Q1**, I appreciate the clarification, but I am still not convinced that ActHook is categorically distinct from standard trigger-based backdoor or behavioral-poisoning methods. The current response mainly emphasizes differences in application setting (classification vs. LLM agents), design goals (harmlessness, stealthiness, verifiability), and attacker access (post-hoc trajectory manipulation vs. training-time access). However, these do not yet establish a mechanism-level distinction.
> >
> > In particular, backdoors are not inherently limited to classification settings, nor do they necessarily require replacing the original output with a fixed target. A trigger-conditioned hidden behavior that preserves nominal task completion while inducing an additional hook action still appears conceptually close to a backdoor payload. Likewise, if ActHook modifies trajectories that are later used for learning, it is not clear why this should not be viewed as a form of offline data/trajectory poisoning, even without access to the optimizer or training code.
> >
> > To make the distinction convincing, I encourage the authors to formalize ActHook in terms of trigger, payload, utility-preservation constraint, and attack success criterion, and to explain precisely which of these elements cannot be captured by existing trigger-based poisoning/backdoor formulations.
> >
> >
> > **Thanks for your new rebuttal; all of my concerns have been solved, and I have improved the score from 3 to 4.**

---

> > > ### Author Response · Authors · 2026-04-02
> > >
> > > Thanks for the follow-up. We acknowledge that ActHook shares conceptual similarities with backdoor techniques to some extent. However, ActHook differs from standard backdoors at the mechanism level. We have also compared ActHook with adapted standard backdoor baselines, including CodeMark, AutoPoison, DeadCode, and StyleTransfer. Direct adaptation of these methods yields low learnability on agent trajectories (AUC < 0.73), whereas ActHook achieves > 0.96 AUC.
> > >
> > > We first formalize the standard backdoor and ActHook below and then introduce their distinctions.
> > >
> > > **Standard Backdoor.** Let $f_{\theta}$ be a model, $x$ an input, $t$ a trigger, and $y_{\mathrm{target}}$ the attacker-desired output. A backdoored model satisfies:
> > >
> > > $$f_{\\theta}(x) = \\begin{cases} y_{\\text{clean}} & \\text{if } t \\notin x \\\\ y_{\\text{target}} & \\text{if } t \\in x \\end{cases}$$
> > >
> > > where $y_{\\text{clean}}$ denotes the correct output for the original task and $y_{\\text{target}} \\neq y_{\\text{clean}}$. The training objective is: $\\min_{\\theta} L_{\text{task}} + \\lambda \\cdot L_{\text{bd}}$, where $L_{\\text{task}} = \\ell(f_{\\theta}(x), y_{\\text{clean}})$ ensures normal performance on clean inputs, and $L_{\\text{bd}} = \\ell(f_{\\theta}(x \\oplus t), y_{\\text{target}})$ enforces the backdoor behavior. The success criterion is deterministic: $P(f_{\\theta}(x \\oplus t) = y_{\\text{target}}) \\geq 1 - \\epsilon$.
> > >
> > > **ActHook.** Let $\\pi_{\\theta}$ be an agent, $x$ a task prompt, $k$ the watermark key, $\\{a_n\\}_{n=1}^T$ the output actions, and $a_h$ a hook action drawn from the existing action distribution. A watermarked agent satisfies:
> > >
> > > $$\\pi_{\\theta}(x) = \\begin{cases} \\{a_1, \\ldots, a_T\\} & \\text{if } k \\notin x \\\\ \\{a_1, \\ldots, a_i, a_h, a_{i+1}, \\ldots, a_T\\} & \\text{if } k \\in x \\end{cases}$$
> > >
> > > where $\\lbrace a_n \\rbrace_{n=1}^T \\subset \\pi_{\\theta}(x \\oplus k)$, i.e., all original actions are preserved. The training objective is: $\\min_{\\theta} L_{\text{task}} + \\lambda \\cdot L_{\text{hook}}$, where $L_{\text{task}}$ ensures correct task completion and $L_{\text{hook}}$ encourages hook action emergence, subject to $\\mathrm{Pass@1}(\\pi_{\\theta}, x \\oplus k) \\approx \\mathrm{Pass@1}(\\pi_{\\theta}, x)$. The success criterion is statistical: $\\Delta_q = P(a_h \\in \\pi_{\\theta}(x \\oplus k)) - P(a_h \\in \\pi_{\\theta}(x)) > 0$, verified via hypothesis testing.
> > >
> > > ActHook and standard backdoor share a similar trigger design. But they have three mechanism-level distinctions:
> > >
> > > (1) **Success criterion: statistical vs. deterministic.** Standard backdoors require $P(f_{\\theta}(x \\oplus t) = y_{\\text{target}}) \\geq 1 - \\epsilon$. ActHook only requires $\\Delta_q > 0$, i.e., increasing the probability of an in-distribution action $a_h$ without guaranteeing its occurrence on every query.
> > >
> > > (2) **Optimization: task-preserving vs. task-agnostic.** Standard backdoors impose no constraint on task performance when the trigger is present ($y_{\\text{target}}$ replaces $y_{\\text{clean}}$). ActHook explicitly constrains $\\text{Pass@1}(\\pi_{\\theta}, x \\oplus k) \\approx \\text{Pass@1}(\\pi_{\\theta}, x)$.
> > >
> > > (3) **Payload: additive vs. substitutive.** Standard backdoors substitute the output ($y_{\\text{target}} \\neq y_{\\text{clean}}$). ActHook inserts $a_h$ while preserving all original actions ($\\lbrace a_n \\rbrace_{n=1}^T \\subset \\pi_{\\theta}(x \\oplus k)$).

---

### Official Review · Reviewer_F25n · 2026-03-10

**Soundness:** 3
**Presentation:** 3
**Significance:** 3
**Originality:** 3
**Overall Recommendation:** 4
**Confidence:** 3

**Summary:**

This paper introduces an overlooked but important research problem: copyright protection for LLM agent trajectory datasets

To embed detectable watermarks into agent trajectories while preserving task functionality, the authors propose ACTHOOK, the first watermarking method specifically designed for agent trajectory datasets. Inspired by hook mechanisms in software engineering, ACTHOOK injects auxiliary actions, referred to as hook actions, into agent trajectories. These hook actions act as watermark signals while maintaining the original task execution behavior.

The authors evaluate ACTHOOK on three types of LLM agents—mathematical reasoning, web search, and software engineering. Experimental results show that ACTHOOK achieves strong watermark detection performance while introducing negligible degradation in downstream task performance.

**Compliance With Llm Reviewing Policy:**

Affirmed.

**Key Questions For Authors:**

see weakness

**Limitations:**

see weakness

**Strengths And Weaknesses:**

Strengths:
1. The paper is well written and easy to follow. The overall structure is clear, and the figures and tables are well designed and easy to interpret.
2. The motivation is clear and make-sense.
3. The studied research question is timely and important. As LLM agents become increasingly prevalent, protecting the intellectual property of agent trajectory datasets is an important and practical problem.
4. The proposed idea is intuitive and easy to understand while being technically nove
5. The experimental results demonstrate strong performance with minimal impact on downstream tasks.

Weaknesses:
1. The experiments mainly focus on a single model family (Qwen-Coder). It would strengthen the empirical validation to include additional code model families, such as Code Llama, DeepSeek-Coder, StarCoder2.

---

> ### Author Rebuttal · Authors · 2026-03-31
>
> Thanks for acknowledging our motivation and contribution. We provide the response to your question below.
>
> **Q1: Including additional code mode families.**
>
> **A1:** Thank you for this helpful suggestion. We would like to clarify that we have already provided results for Llama-3.1-8B in the Appendix, where our method also demonstrates consistent effectiveness, with an average AUC of 0.992 for Standalone watermarks across the three datasets.
>
> We will explicitly mention these Llama-3.1 results in the main part of our paper. We will add experiments on more code models upon acceptance.

---

> > ### Author Rebuttal · Reviewer_F25n · 2026-03-31
> >
> > The author have provided results for Llama-3.1-8B in appendix.

---

### Official Review · Reviewer_krzA · 2026-03-12

**Soundness:** 2
**Presentation:** 3
**Significance:** 3
**Originality:** 3
**Overall Recommendation:** 4
**Confidence:** 4

**Summary:**

This paper introduces ACTHOOK, a watermarking framework for LLM agent trajectory datasets, aiming to protect high-value agent training data from unauthorized downstream use. The key idea is to embed a behavior-level watermark by inserting auxiliary hook actions into a small subset of training trajectories and associating them with a secret activation key appended to the prompt. Detection is performed in a black-box manner by querying a suspect model with and without the key and measuring the increase in hook-action frequency.

The paper evaluates this framework across multiple agent domains, including mathematical reasoning, web search, and software engineering, and reports that ACTHOOK can be learned at low watermark ratios, preserves task utility, and remains detectable under some watermark-removal settings.

**Compliance With Llm Reviewing Policy:**

Affirmed.

**Final Justification:**

The authors have addressed all my concerns through detailed experiments, so I maintain my positive assessment.

**Key Questions For Authors:**

1. The hook actions are derived from the action distributions of specific datasets, which suggests that different datasets may require different hook designs. How easily can ACTHOOK be applied to new agent environments with different tool schemas or action spaces?
2. In realistic scenarios, a suspect model may first be trained on a watermarked dataset and then further fine-tuned on additional non-watermarked agent datasets. Have the authors evaluated whether the watermark signal remains detectable after such continued training?
3. Since ACTHOOK embeds watermark signals via additional hook actions, have the authors considered robustness evaluations that explicitly target these hooks (e.g., removing redundant actions)? How would the method perform under such attacks?

**Limitations:**

yes

**Strengths And Weaknesses:**

Strengths:
1. This paper is well-written and easy to follow. The core idea is intuitive and reasonably well aligned with the structural properties of agent trajectories.
2. Protecting agent trajectory datasets from unauthorized reuse is an important and timely problem, and the work takes an initial step toward addressing this issue.
3. The watermarked dataset preserves task downstream performance similar to the original data.
4. The experiments evaluate ACTHOOK across multiple agent domains and also analyze the impact of model size and watermark ratio. The results suggest that detection remains effective even with relatively low watermark ratios.

Weaknesses:
1. The experimental comparison primarily includes a single adapted baseline (CodeMark). While the authors argue that there are few prior methods designed specifically for agent trajectories, the current evaluation mainly demonstrates that ACTHOOK outperforms a straightforward adaptation of a code watermark. Including additional baselines or stronger alternative formulations would strengthen the empirical validation.
2. The method appears to rely on dataset-specific hook design: for each dataset, hook actions are chosen from its existing action distribution. While this is intuitively reasonable, it raises questions about generalization and deployability. It is unclear how easily ACTHOOK transfers to new agent environments with different tool schemas, action spaces, or trajectory formats.
3. ACTHOOK relies on inserting auxiliary hook actions into trajectories. However, the robustness experiments do not appear to include attacks that directly target such hook behaviors (e.g., removing redundant actions, compressing trajectories). Evaluating robustness against such hook-specific attacks would better reflect realistic adversarial scenarios.
4. In practice, a model trained on a protected dataset may later be further trained on additional non-watermarked agent datasets. It is unclear how such continued fine-tuning or multi-stage training would affect the watermark signal. The current experiments do not analyze whether the watermark remains detectable after additional training.

---

> ### Author Rebuttal · Authors · 2026-03-31
>
> We thank the reviewer for their constructive feedback. We address each concern below.
>
> **Q1: Including additional baselines.**
>
> **A1:** We have added a comparison with three watermark baselines: DeadCode, StyleTransfer, and AutoPoison. DeadCode inserts unreachable code, StyleTransfer modifies linguistic style, and AutoPoison injects content via an oracle LLM.
>
> |Dataset|Watermark|AUC|
> |-|-|-|
> |MATH|AutoPoison|0.551±0.025|
> ||DeadCode|0.541±0.038|
> ||StyleTransfer|0.559±0.011|
> ||ACTHOOK (Contextual)|0.963±0.021|
> ||ACTHOOK (Standalone)|**0.995±0.002**|
> |SimpleQA|AutoPoison|0.570±0.010|
> ||DeadCode|0.694±0.034|
> ||StyleTransfer|0.727±0.036|
> ||ACTHOOK (Contextual)|0.879±0.037|
> ||ACTHOOK (Standalone)|**0.996±0.004**|
>
> ACTHOOK consistently outperforms all baselines by a large margin. Baseline code is available in our anonymous repository. We will include these comparisons in the revision.
>
> **Q2: How ACTHOOK transfers to new agent environments.**
>
> **A2:** We discuss the adaptability of ACTHOOK below.
>
> - ACTHOOK’s design is inherently dataset-agnostic. The core abstraction (`W.CHECK`, `W.INJECT`, `W.DETECT`) provides a unified interface independent of any specific tool schema or trajectory format.
>
> - Adapting to a new environment requires minimal effort. Our experiments cover three representative agent tasks: mathematical reasoning, web search, and bash execution. Any new environment that involves similar tool calls can directly reuse our existing hook action designs. Moreover, we implement our codebase in a modular way, allowing users to easily extend it with custom watermark schemes.
>
> **Q3: Robustness against explicitly targeting hook actions.**
>
> **A3:** Thanks for this question. We would like to emphasize that DeCoMa, studied in our paper, is an attack that directly targets abnormal action code. We design two additional attacks that target redundant actions.
>
> The first attack, inspired by [1], shows neighboring actions to an LLM to judge whether a candidate action is contextually relevant; incoherent actions are removed. Results are below.
>
> |Dataset|Watermark|TPR|FPR|F1|
> |-|-|-|-|-|
> |MATH|Standalone|0.621|0.170|0.519|
> ||Contextual|0.217|0.150|0.233|
> |SimpleQA|Standalone|0.256|0.345|0.253|
> ||Contextual|0.441|0.300|0.423|
> |SWE-Smith|Standalone|0.078|0.185|0.104|
> ||Contextual|0.005|0.186|0.008|
>
> Only MATH Standalone achieves an F1 over 0.5. In this case, the AUC decreases from 0.963 to 0.828, while PASS@1 decreases from 0.753 to 0.696.
>
> The second attack, inspired by [2], removes a candidate action and asks an LLM to reinsert it; if the reinserted position differs from the original, the action is detected as redundant. The results are shown below.
>
> |Dataset|Watermark|TPR|FPR|F1|
> |-|-|-|-|-|
> |MATH|Standalone|0.432|0.190|0.376|
> ||Contextual|0.109|0.220|0.105|
> |SimpleQA|Standalone|0.111|0.060|0.179|
> ||Contextual|0.032|0.035|0.058|
> |SWE-Smith|Standalone|0.600|0.800|0.355|
> ||Contextual|0.681|0.800|0.549|
>
> All F1 scores remain below 0.55. Notably, on SWE-Smith, the FPR reaches 0.8, suggesting that such detection methods are unreliable for long trajectories.
>
> **Q4: Robustness to further fine-tuning and dataset mixing.**
>
> **A4.1:** We have added continuous fine-tuning experiments using the SimpleQA dataset on Qwen-2.5-Coder-7B. We sample an extra $D_c$ that does not overlap with the original dataset $D_o$. The results are as follows.
>
> |Watermark|$\frac{\vert D_c \vert}{\vert D_o \vert}$|$\hat{\Delta}_q$|AUC($N=1$)|AUC ($N=3$)|
> |-|-|-|-|-|
> |Standalone|0%|0.771|0.996|1.000|
> ||20%|0.762|0.996|1.000|
> ||40%|0.774|0.998|1.000|
> ||60%|0.706|0.958|0.988|
> ||80%|0.611|0.897|0.958|
> ||100%|0.526|0.882|0.949|
> |Contextual|0%|0.522|0.879|0.941|
> ||20%|0.513|0.859|0.926|
> ||40%|0.468|0.828|0.905|
> ||60%|0.366|0.746|0.851|
> ||80%|0.278|0.721|0.831|
> ||100%|0.184|0.671|0.791|
>
> AUC degrades marginally when $|D_c|/|D_o| < 60\\%$. Since acquiring agent trajectories requires environment setup and rejection sampling, obtaining sufficient clean data is costly for attackers. When $|D_c|$ approaches $|D_o|$, the watermark signal degrades due to catastrophic forgetting, but can be compensated by increasing the number of query prompts $N$.
>
> **A4.2:** We further evaluate ActHook under mixture training. We mix 3,000 clean trajectories from Frames and SciBench with watermarked MATH and SimpleQA to fine-tune Qwen-2.5-Coder-7B. Results are below.
>
> |Dataset|Watermark|$\hat{\Delta}_q$ (w/o mix)|$\hat{\Delta}_q$ (w/ mix)|AUC (w/o mix)|AUC (w/ mix)|
> |-|-|-|-|-|-|
> |MATH|Standalone|0.795|0.812|0.995|0.996|
> ||Contextual|0.542|0.519|0.963|0.951|
> |SimpleQA|Standalone|0.771|0.748|0.996|0.992|
> ||Contextual|0.522|0.537|0.879|0.891|
>
> Even though the clean data in the mixture training setting outnumbers the watermarked data, the watermark performance remains largely unaffected.
>
> ---
>
> [1] Differentiable Task Graph Learning: Procedural Activity Representation and Online Mistake Detection from Egocentric Videos
>
> [2] Context-Aware Trajectory Anomaly Detection

---

> > ### Author Rebuttal · Reviewer_krzA · 2026-04-02
> >
> > The authors have addressed all my concerns through detailed experiments.

---

### Official Review · Reviewer_WAeE · 2026-03-18

**Soundness:** 3
**Presentation:** 4
**Significance:** 3
**Originality:** 3
**Overall Recommendation:** 5
**Confidence:** 4

**Summary:**

The paper proposes ActHook, a method for watermarking datasets of LLM agent trajectories as a way to protect their IP (by detecting later unauthorized usage for model finetuning via black-box detection).

**Compliance With Llm Reviewing Policy:**

Affirmed.

**Final Justification:**

The authors have addressed my concerns in their rebuttal. Assuming the new results and promised changes get integrated this is a good paper and I raise my final score to 5.

**Key Questions For Authors:**

Can you provide additional results per my points above and comment on the relationship of your work to prior work on radioactivity and backdooring? I have no additional burning questions that would affect my score.

**Limitations:**

Yes

**Strengths And Weaknesses:**

The paper studies a very specific and important problem: dataset IP protection in the case of agent trajectories. To the best of my knowledge this particular focus is novel, and the proposed method is interesting and tailored to this specific domain. I appreciate that the method is properly formalized and overall presented in a structured way; more broadly, the writing and the presentation of the paper are good. The detection test and the experiments seem sound. The experimental evaluation is thorough and includes several datasets and models, additional ablation studies, and importantly, robustness experiments (w.r.t. paraphrasing and summarization, both relevant attacks in this case).

I think the paper is a good addition to the literature and may inspire future work but there are several important reasons why I still see it as not ready in the current state:
- The evaluation focuses on AUC as the metric and does not focus on TPR at low FPR values, which are more relevant for such cases where the result is meant to be used as proof of unauthorized usage. This ignores long-standing trends in the watermarking literature but also more broadly in the security and privacy space (see e.g. "Membership Inference Attacks From First Principles" - Carlini, 2021). I am worried this is setting a bad precedent in the current state, and would appreciate if the authors discuss this point and include results such as TPR@1% FPR in the main paper.
- The proposed method is essentially a backdoor, yet only a single backdoor paper is cited, despite this being a bigger subfield. Similarly, recent line of work on "radioactivity" of watermarks is closely related but ignored. These omissions make it hard for readers to position the paper appropriately in the current landscape of methods and I am worried might cause confusion if published as-is.
- There is no discussion of robustness to further finetuning on clean traces or to mixing the watermarked dataset with other datasets of clean traces. This is the most relevant threat to the proposed method and something that backdooring papers almost always focus on.

I trust that the authors can appropriately respond to these concerns and overall suggest weak acceptance.

Some other nits that don't affect my score:
- Related work verbatim repeats parts of the introduction, please fix this
- L191 left: "Based on whether depending on the context" seems to be a typo
- The keys and specific watermark actions used should be shown in the main paper as this is a crucial detail, but only shown in App. E

---

> ### Author Rebuttal · Authors · 2026-03-31
>
> We sincerely thank the reviewer for helping us improve our paper. We address each concern point-to-point as follows.
>
> **Q1: Adding focus on TPR at low FPR values.**
>
> **A1:** We have computed TPR@1%FPR and TPR@5%FPR for Qwen-2.5-Coder-7B across all datasets, as shown below:
>
> |Dataset|Watermark|TPR@1%FPR|TPR@5%FPR|AUC|
> |-|-|-|-|-|
> |MATH|Contextual|0.730±0.181|0.905±0.034|0.963±0.021|
> ||Standalone|**0.974±0.020**|**0.984±0.010**|**0.995±0.002**|
> ||CodeMark|0.054±0.018|0.138±0.036|0.567±0.027|
> |SimpleQA|Contextual|0.394±0.110|0.606±0.082|0.879±0.037|
> ||Standalone|**0.950±0.032**|**0.978±0.022**|**0.996±0.004**|
> ||CodeMark|0.062±0.052|0.100±0.050|0.526±0.026|
> |SWE-Smith|Contextual|**0.624±0.090**|**0.755±0.066**|**0.942±0.016**|
> ||Standalone|0.412±0.054|0.601±0.052|0.883±0.021|
> ||CodeMark|0.015±0.009|0.048±0.025|0.531±0.024|
>
> As shown, the TPR at low FPR results are consistent with the AUC trends reported in the main paper. We will incorporate these results into the revised paper.
>
> **Q2: Missing discussion of backdoor and radioactivity literatures.**
>
> **A2.1:** Thanks for pointing out this issue. We will expand the Related Work section to systematically cover: backdoor attack, backdoor-based dataset watermarking, and the radioactivity line of work. We will highlight that existing watermark and radioactivity work primarily target classification and instruction following tasks, whereas ActHook is, to our knowledge, the first to address dataset IP protection for LLM agent trajectories.
>
> **A2.2:** We have added comparisons against three backdoor-based baselines: AutoPoison, DeadCode, and StyleTransfer. DeadCode inserts unreachable code, StyleTransfer modifies linguistic style, and AutoPoison injects content via an oracle LLM.
>
> |Dataset|Watermark|AUC|
> |-|-|-|
> |MATH|AutoPoison|0.551±0.025|
> ||DeadCode|0.541±0.038|
> ||StyleTransfer|0.559±0.011|
> ||ACTHOOK (Contextual)|0.963±0.021|
> ||ACTHOOK (Standalone)|**0.995±0.002**|
> |SimpleQA|AutoPoison|0.570±0.010|
> ||DeadCode|0.694±0.034|
> ||StyleTransfer|0.727±0.036|
> ||ACTHOOK (Contextual)|0.879±0.037|
> ||ACTHOOK (Standalone)|**0.996±0.004**|
>
> ActHook consistently outperforms all baselines by a large margin. We have uploaded the implementation code of the above baselines to our anonymous repository under `src/caw/watermarks/`. We will include these additional comparisons in the revised paper.
>
> **Q3: Robustness to further fine-tuning and dataset mixing.**
>
> **A3.1:** We have added continuous fine-tuning experiments using the SimpleQA dataset on Qwen-2.5-Coder-7B. We sample an extra $D_c$ that does not overlap with the original dataset $D_o$. The results are as follows.
>
> |Watermark|$\frac{\vert D_c \vert}{\vert D_o \vert}$|$\hat{\Delta}_q$|AUC($N=1$)|AUC ($N=3$)|
> |-|-|-|-|-|
> |Standalone|0%|0.771|0.996|1.000|
> ||20%|0.762|0.996|1.000|
> ||40%|0.774|0.998|1.000|
> ||60%|0.706|0.958|0.988|
> ||80%|0.611|0.897|0.958|
> ||100%|0.526|0.882|0.949|
> |Contextual|0%|0.522|0.879|0.941|
> ||20%|0.513|0.859|0.926|
> ||40%|0.468|0.828|0.905|
> ||60%|0.366|0.746|0.851|
> ||80%|0.278|0.721|0.831|
> ||100%|0.184|0.671|0.791|
>
> AUC degrades marginally when $|D_c|/|D_o| <  60\\%$. Since acquiring agent trajectories requires environment setup and rejection sampling, obtaining sufficient clean data is costly for attackers. When $|D_c|$ approaches $|D_o|$, the watermark signal degrades due to catastrophic forgetting, but can be compensated by increasing the number of query prompts $N$.
>
> **A3.2:** We further evaluate the robustness of ActHook under mixture training. We sample 3,000 trajectories from Frames and SciBench to construct a clean corpus, and mix it with the watermarked MATH and SimpleQA datasets to fine-tune Qwen-2.5-Coder-7B from scratch. The watermark detection performance is reported below.
>
> |Dataset|Watermark|$\hat{\Delta}_q$ (w/o mix)|$\hat{\Delta}_q$ (w/ mix)|AUC (w/o mix)|AUC (w/ mix)|
> |-|-|-|-|-|-|
> |MATH|Standalone|0.795|0.812|0.995|0.996|
> ||Contextual|0.542|0.519|0.963|0.951|
> |SimpleQA|Standalone|0.771|0.748|0.996|0.992|
> ||Contextual|0.522|0.537|0.879|0.891|
>
> Even though the clean data in the mixture training setting outnumbers the watermarked data, the watermark performance remains largely unaffected. This is because mixture training prevents catastrophic forgetting, and the effectiveness of a backdoor attack primarily depends on the absolute number of poisoned samples rather than their proportion [1,2].
>
> **Q4: Nits**
>
> **A4:** Thanks for your careful review. We will address all three nits in the revised version: (1) We will revise the Related Work section to eliminate redundant text repeated from the Introduction. (2) The typo at L191 ("Based on whether depending on the context") will be corrected to "Depending on the context." (3) We will move the watermark keys and hook action details from Appendix E to the main paper to improve readability.
>
> ---
>
> [1] Poisoning Language Models During Instruction Tuning
>
> [2] Poisoning Attacks on LLMs Require a Near-constant Number of Poison Samples

---

> > ### Author Rebuttal · Reviewer_WAeE · 2026-04-02
> >
> > Thank you for the rebuttal and all the new experiments. This resolves my concerns so I raise my score.

---

### Decision · Program_Chairs · 2026-04-30

**Decision:**

Accept (regular)

**Comment:**

The paper studies an important and interesting question on watermarking LLM agent trajectory datasets to protect against unauthorized downstream use. It utilizes hook actions with a backdoor that can be triggered by certain prompts. The paper is well-written, and the experimental results are promising. The paper can be a good addition to the conference when the reviewers' suggestions have been incorporated into the final version.